# Hippocampal-prefrontal connectivity relates to inter-individual differences and training gains in distinguishing similar memories

Panagiotis Iliopoulos [1,2,6] ✉, Jeremie Güsten[1,6], Eóin Molloy[2,3,4], Radoslaw Martin Cichy [5], Friedrich Krohn[1,2], Anne Maass[2] & Emrah Düzel[1,2]

Mnemonic discrimination (MD) is the ability to distinguish current experiences from similar memories. Research on the brain correlates of MD has focused on how regional neural responses are linked to MD. Here we go beyond this approach to investigate inter-regional functional connectivity patterns related to MD, its inter-individual variability and training-related improvement. Based on prior work we focused on medial temporal lobe (MTL), prefrontal cortex (PFC) and visual regions. We used fMRI to determine how connectivity patterns between these regions are related to MD before and after 2-weeks of web-based cognitive training. We found MD-related connectivity involving MTL-PFC-visual areas. Hippocampal-PFC connectivity was negatively associated with inter-individual variability in MD performance across two tasks. Hippocampal-PFC connectivity decrease was also linked to inter-individual variability in post-training MD improvement. Additionally, training led to increased connectivity from the lateral occipital cortex to the occipital pole area. Our results point to a hippocampal-PFC connectivity pattern that is a reliable marker of MD performance. This pattern is further related to MD training gains providing strong evidence for its role in distinguishing similar memories. Overall, we show that hippocampal-PFC connectivity constitutes a neural resource for MD that enables training-related improvement and may be targeted to enhance cognition.

The ability to remember and distinguish between similar experiences is fundamental to human cognition. This capacity, known as mnemonic discrimination (MD), enables us to form distinct memories. MD is putatively supported by a process called pattern separation, whereby the brain orthogonalizes overlapping neural representations to prevent interference between similar memories[1,2]. MD and pattern separation are predominantly attributed to the hippocampus and associated medial temporal lobe (MTL) structures[1–3]. The clinical relevance of MD is illustrated by its decline with aging and Alzheimer's disease (AD) pathology[1,4–7]. Older adults have intact recognition for exact repetitions of previously encountered stimuli ("repeats"), but they show deficits in identifying similar versions of previously encountered stimuli ("lures")[1,7]. This deficit in distinguishing between similar memories is even stronger in individuals with AD, leading

to memory confusion and errors[1]. There is, therefore, a strong interest in understanding the neural correlates of MD, the neural basis of its inter-individual variability and its improvement by training; not only for elucidating fundamental memory processes but also for developing interventions to mitigate cognitive decline.

In humans, functional magnetic resonance imaging (fMRI) has been instrumental in advancing our knowledge of the neural networks involved in MD. By examining the differential brain activation elicited by similar (lure) versus repeated stimuli—a contrast known as lure detection (LD)—researchers have identified regions implicated in MD and by inference also in pattern separation[5,8–10]. The LD contrast compares brain responses to correctly identified lures with responses to repeated items that require no discrimination, thereby indexing discrimination-related responses relative

[1]Institute of Cognitive Neurology and Dementia Research, Otto-von-Guericke University, Magdeburg, Germany. [2]German Center for Neurodegenerative Diseases (DZNE), Magdeburg, Germany. [3]Division of Nuclear Medicine, Department of Radiology & Nuclear Medicine, Faculty of Medicine, Otto von Guericke University Magdeburg, Magdeburg, Germany. [4]AICURA medical GmbH, Colditzstraße 34/36, 16A, Berlin, Germany. [5]Department of Education and Psychology, Freie Universität Berlin, Berlin, Germany. [6]These authors contributed equally: Panagiotis Iliopoulos, Jeremie Güsten. ✉e-mail: panagiotis.iliopoulos@med.ovgu.de

to a repetition baseline. This contrast is well-established in MD paradigms[5,8–10] and has been linked to both MD performance and biomarker data[9]. Findings from fMRI studies indicate that, in addition to the hippocampus/dentate gyrus[2,11], a broader network of brain regions is implicated in MD[12,13]. These include other MTL regions[14], the prefrontal cortex (PFC), and areas in the visual cortex[12,13,15–17].

Here we addressed the question of how MTL, PFC and visual areas functionally interact to support MD. To that end, we took a two-pronged approach. First, we determined how functional connectivity between these regions during an MD task related to inter-individual variability in performance across two different MD paradigms. Second, we performed an MD training intervention and determined whether training-related gains in MD performance were related to the same functional connectivity patterns that also explained inter-individual variability. Our rationale is that the convergence between functional connectivity related to inter-individual variability and training-related gains can uncover neural resources that underlie MD more robustly than the cross-sectional and univariate studies to date.

Our study builds on previous work showing that it is possible to enhance MD performance with cognitive training. A recent study performed in our lab reported decreased false recognition of lure stimuli and regional brain activation changes following a 2-week web-based cognitive intervention[10]. Here, we determined whether cognitive training altered functional connectivity within the MD network and whether these changes resembled the patterns displayed in high-performing individuals. We also tested whether such a connectivity pattern relates to performance in another MD task[18]. An overarching question is whether neural resources explaining inter-individual variability also explain training-related gains. If we find convergence -the same neural networks are involved in both individual differences and training gains- it would indicate that high performers and beneficiaries of cognitive training tap into similar neural resources. This would indicate that these networks are fundamental to MD and strengthen our confidence in targeting them in future research. Conversely, if there is no convergence, it would imply that training induces neural adaptations distinct from the observed inter-individual variability patterns. While it is possible that inter-individual differences in MD are related to inter-individual differences in hippocampal circuitry, cognitive training may lead to improvements in MD by modifying afferent and efferent connections of hippocampal circuitry, including the fidelity of visual representations.

We hypothesize that effective MD is supported by enhanced bidirectional connectivity among an MTL-PFC-visual network. Our rationale is that such connectivity would facilitate the integration of visual information with memory processes. We expect specifically hippocampal-PFC interactions to play a central role, given findings indicating a role of PFC, alongside the hippocampus, in MD both in humans[12,13,16,17,19] and animal studies[20]. We further hypothesize that training will lead to altered connectivity within the MD-associated network, reflecting neural plasticity in the form of functional reorganization[21,22]. We expect the connectivity changes to converge with the connectivity pattern related to inter-individual performance differences.

For these objectives, we employed a two-week web-based cognitive training protocol emphasizing MD, using a 2×2 longitudinal intervention design (Group × Time)[10]. Functional connectivity was assessed through region-of-interest generalized psychophysiological interaction (gPPI) analyses[23], allowing us to model event-related functional connectivity between brain regions while accounting for task-related activity. Participants underwent pre- and post-training fMRI scanning at 3 Tesla, performing an established MD task, which includes both object and scene stimuli[4,8]. Our primary outcome measure was the functional connectivity associated with correct lure detection compared to repetition trials. Additionally, the participants completed a battery of behavioral paradigms that included another established MD task, the mnemonic similarity task (MST)[18]. We hypothesize that the connectivity pattern related to inter-individual differences will be linked to performance measured in this different MD task.

By integrating analyses of functional connectivity with cognitive training, our study aims to advance the understanding of the neural mechanisms underlying MD and their potential plasticity. This knowledge could inform the development of targeted interventions to enhance memory function and contribute to preventative strategies against cognitive decline in aging and neurodegenerative diseases.

We find that MD is associated with functional interactions across medial temporal, prefrontal, and visual cortices. Stronger hippocampal–prefrontal coupling is negatively associated with MD performance across two tasks, and individuals who improve most with cognitive training show the greatest reductions in this coupling. Training also increases MD-related connectivity from the lateral occipital cortex to the occipital pole. Our results identify hippocampal–prefrontal connectivity as a reliable, behaviorally relevant marker of MD and suggest it may be a modifiable neural resource that can be targeted to enhance the discrimination of similar memories.

## Results

To assess the impact of cognitive training on mnemonic discrimination (MD) and its neural mechanisms, we analyzed behavioral and fMRI data from 54 healthy young adults who completed pre- and post-intervention sessions. Participants were randomly assigned to either the experimental training group ($n = 26$), which received MD-focused cognitive training, or the active control group ($n = 27$), which engaged in a psychomotor task without an MD component (Fig. 1). During fMRI scanning sessions, all participants performed a six-back object-scene continuous recognition task (response options: "old" or "new") that required distinguishing between highly similar ("lure") and identical ("old") images, engaging MD (Fig. 1A–C). The task consisted of two runs, each comprising 60 experimental trials -divided among lure and repeat conditions for both object and scene trial types- and 20 baseline perceptual control trials. Detailed descriptions of the participant groups, behavioral tasks and training protocol are provided in the Methods section and in ref. 10. For an overview of the analysis flow, see Fig. 2.

### Baseline (pre-training) connectivity results

**Lure detection connectivity profile spans MTL, PFC, and visual cortex.** To first characterize the connectivity pattern during successful MD, we contrasted the event-related connectivity for the correct lure versus repeat trials. We term this difference the lure detection contrast (LD), and it is used for all connectivity analyses in our study. We performed gPPI ROI-to-ROI analysis in the pre-training whole-sample data, using an a priori selection of 11 ROIs covering major prefrontal (PFC), medial temporal lobe (MTL), and visual areas, resulting in an $11 \times 11$ connectivity matrix (110 unique connections). To identify significant patterns of connectivity, we applied cluster-based inference using ROI-to-ROI network multivariate parametric statistics. This approach groups related connections into clusters based on their statistical association, and computes an F-statistic for each cluster. Significance was determined at the cluster level using FDR correction ($p < 0.05$, $p$-FDR). Within each significant cluster, a post-hoc connection-level threshold of $p < 0.05$ (uncorrected) was applied to retain only the strongest contributing connections. Non-significant clusters and connections that did not meet the above threshold within significant clusters were not retained for further analysis.

In accordance with our hypothesis, we observed three significant clusters ($p < 0.05$ $p$-FDR) (see Fig. 3 and Table 1 for statistics) for the LD contrast. The first cluster displayed lower connectivity in visual-to-visual and visual-MTL connections during correct lure detection, specifically between the lateral occipital cortex (LOC) and occipital pole (OP), and between the OP and perirhinal (PRC), as well as OP and entorhinal cortex (EC). Specifically, connectivity for the repeat trials was higher than in the correct lures, driving this effect. The second cluster exhibited higher LD connectivity in LOC – PFC and inferior frontal gyrus pars triangularis (IFG

**Fig. 1 | Experimental six-back object-scene mnemonic discrimination paradigm and study design.**
**A–C** The six-back object-scene mnemonic discrimination (MD) task was performed in the scanner (fMRI) and remotely on a computerized web-based training platform. Stimuli were presented in sequences of 12 items: six encoding ("first" trials) followed by six test phase images. Participants were asked to respond "new" or "old" to each image using their middle and index fingers. In the test phase, they had to recognize whether the stimulus was similar but slightly changed ("lure" correct response: "new") or an identical repetition ("repeat" correct response: "old") compared to the "first" images. Each sequence consisted of either objects (**A**) or scenes (**B**) only. Lure and repeat stimuli only differed in shape or geometry (**C**). **D** Participants completed the six-back object-scene task inside the MRI scanner. They were scanned pre- and post a 2-week remote web-based cognitive training intervention that consisted of three 45-minute sessions per week. One group (*n* = 26) trained using a web-based version of the above task, whereas an active control group (*n* = 27) was presented with the same images but did a psychomotor task by clicking on moving neuron-icons on top of the images. Note that different sets of images were shown for each fMRI session, and the cognitive training.

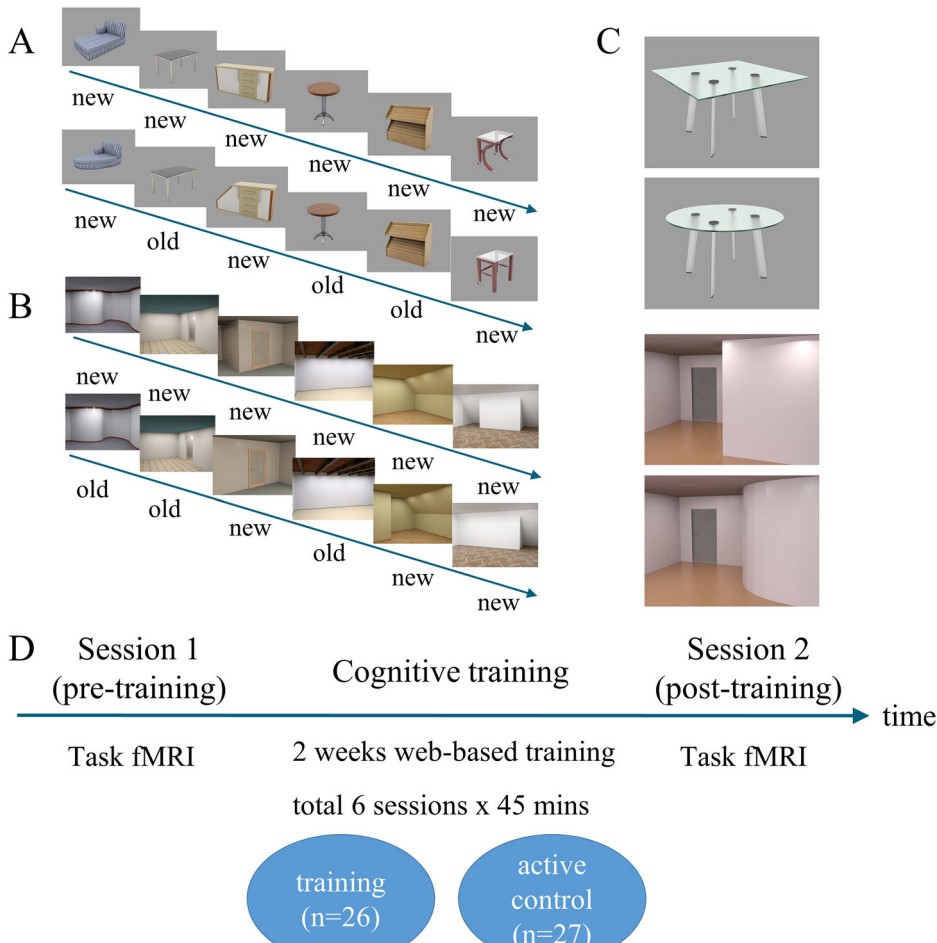

**Fig. 2 | Study analysis flow.** (**1**) To identify the lure detection (LD) connectivity profile (i.e. significant brain connections), we first performed connectivity analyses in the whole-sample pre-training data (*n* = 54). (**2**) Then, focusing on the identified brain connections from the step 1, we fitted linear models to test the link between connectivity in each connection and MD performance at the baseline whole-sample data. (**3**) Next, we compared the LD connectivity for the training versus control group for each connection identified in step 1. Note: One dataset couldnt be analyzed here due to technical error. (**4**) Then, we fitted linear models to test the link between connectivity change and MD performance change (Δ: post-training versus pre-training change); we focused on the brain connections which exhibited a significant link to MD performance at baseline or a training effect (i.e., the significant connections from steps 2 and 3, respectively).

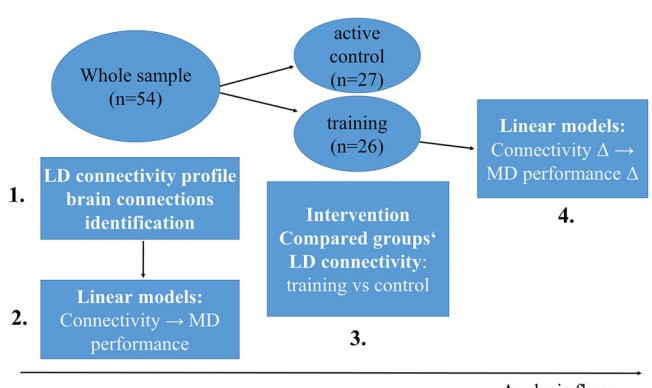

tri)-MTL connections: specifically, between the LOC and the IFG tri, superior frontal gyrus (SFG), and inferior frontal gyrus pars opercularis (IFG oper) areas. In addition, we found higher LD connectivity between the LOC and the hippocampus, and between the IFG tri and the EC, as well as the PRC. The third cluster exhibited higher connectivity during correct lure detection in mainly hippocampal-PFC connections: specifically between the hippocampus and the SFG, IFG oper and IFG tri areas. In summary, we found that MD is associated with lower task-based connectivity in LOC-OP and OP – MTL connections, and higher connectivity in LOC–PFC, LOC-hippocampus, IFG tri-MTL and hippocampal-PFC connections. These clusters reveal a task connectivity profile associated with successful MD that involves specific MTL-PFC-visual connections. Thus, all our subsequent connectivity analyses focused on these identified MD-associated functional networks (Fig. 3A).

## Higher LD connectivity in hippocampal-PFC connections is linked to poorer mnemonic discrimination performance

Following the characterization of baseline MD-associated functional connectivity, we investigated whether there is a quantitative relationship between this LD task connectivity in each cluster and MD behavioral performance (defined using the *A* discriminability index). To test this, we fitted linear regression models between the LD connectivity in each connection and the *A* discriminability index (covariates: age, sex).

Results show that lower LD task connectivity in hippocampal-PFC connections (cluster 3) is associated with higher MD (Table 2), as shown in each of the following connections: SFG-hippocampus, hippocampus-IFG oper, IFG tri-hippocampus, hippocampus-SFG, and hippocampus-IFG tri

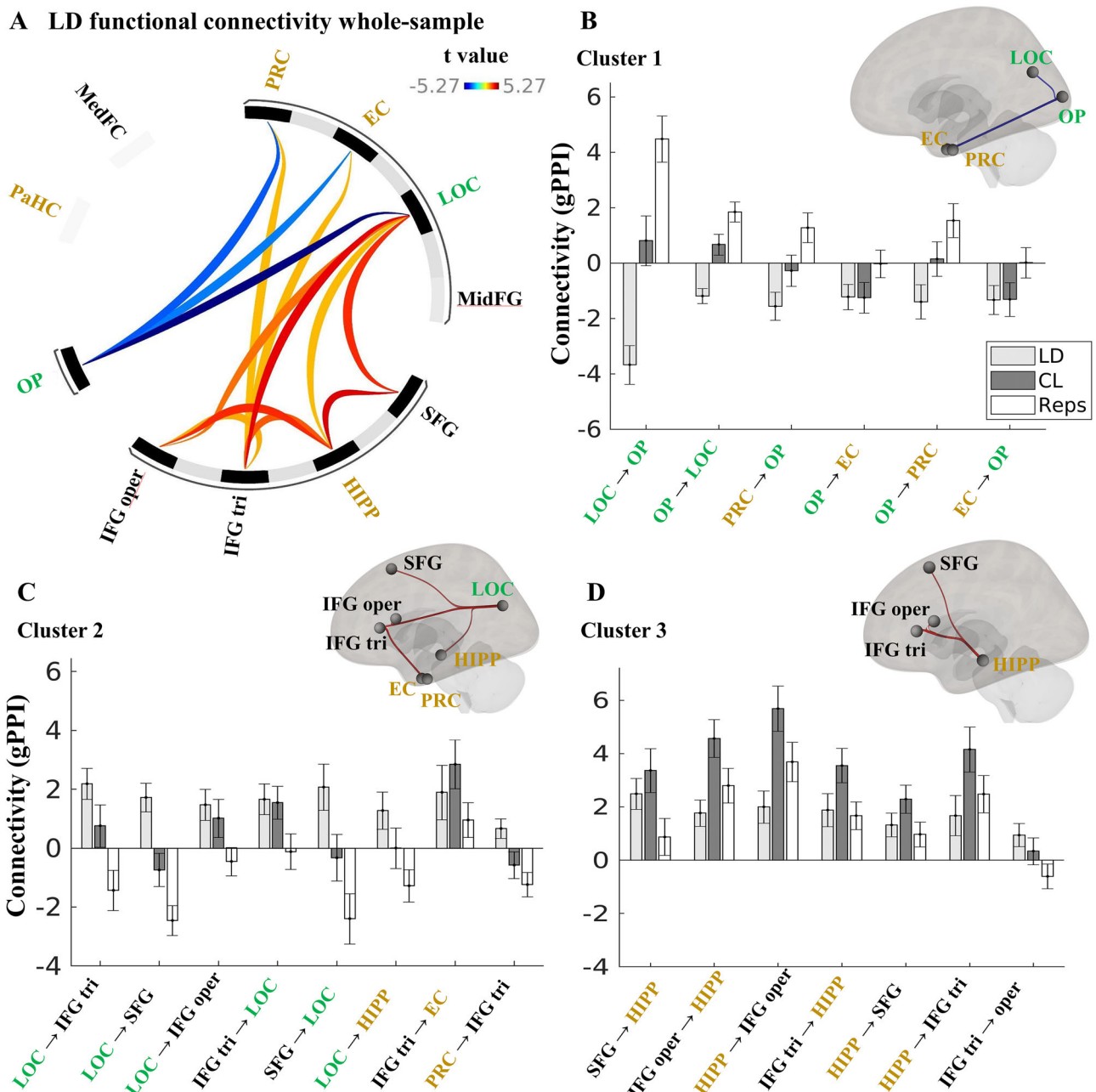

**Fig. 3 | Lure detection (LD) connectivity reveals three clusters spanning MTL, PFC, and visual cortex areas. A** Connectogram displaying significant functional connectivity for the LD contrast (correct lures (CL) versus repeats (Reps) trials). Link colors represent *t*-values (blue indicates reduced connectivity/negative contrast; red/orange indicates increased connectivity/positive contrast). **B**–**D** Detailed profiles for the three identified clusters, including anatomical topography and connection strength (bar plots). The bars show both the LD contrast and each conditions' connectivity. **B** Cluster 1 exhibited lower connectivity (negative LD values) for LOC – OP connections and between the OP and the MTL (EC, PRC) regions. **C** Cluster 2 was characterized by higher connectivity (positive LD values) linking the LOC area with the PFC areas and the IFG tri area to the EC and PRC area. **D** Cluster 3 exhibited

higher connectivity between the hippocampus and the PFC areas (see Table 1 for more detail). Bar plots display the mean gPPI beta estimates (Y axis) for the main LD contrast (light gray) and the separate conditions: Correct lures (CL; dark gray) and repeats (Reps; white), relative to the perceptual baseline control trials. Error bars denote the mean ± the standard error of the mean across participants (*n* = 54). ROI label colors denote anatomical grouping: visual (green), MTL (gold), or PFC (black). PaHC parahippocampal cortex, MedFC medial frontal cortex, PRC perirhinal cortex, EC entorhinal cortex, LOC lateral occipital cortex, MidFG middle frontal gyrus, SFG superior frontal gyrus, HIPP: hippocampus, IFG inferior frontal gyrus; tri pars triangularis, oper pars opercularis. OP occipital pole.

(Table 2 for statistics and Fig. S3). Given that these hippocampal–PFC connections all exhibited a similar negative relationship with MD performance, we created a composite hippocampal–PFC connectivity score by averaging the z-scores of each connection. Using this composite score as the independent variable provided a more concise summary of the overall relationship. This model confirmed the strong negative association between LD connectivity and MD performance (*p* = 0.001, b = −0.032; see Fig. 4).

Conversely, we observed no significant relationship between LD connectivity and MD for any connection within clusters 1 and 2. Within cluster 1, there was a non-significant positive association between OP-LOC connectivity and MD performance (A index) (*p* = 0.055, *p*-FDR = .164, $R^2_p = 0.077$), and a negative link between EC-OP connectivity and MD performance that did not survive correction (*p* = 0.047, *p*-FDR = 0.164, $R^2_p = 0.077$) (Table 2).

**Table 1 | ROI-to-ROI connectivity (gPPI) for the lure detection (LD) contrast**

| Cluster/connection | Statistic | p | p-FDR |
|---|---|---|---|
| Cluster 1 | F(2,52) = 15.15 | <0.001 | <0.001 |
| LOC - OP | T(53) = −5.27 | <0.001 | <0.001 |
| OP - LOC | T(53) = −4.36 | <0.001 | 0.001 |
| PRC-OP | T(53) = −3.05 | 0.004 | 0.036 |
| OP - EC | T(53) = −2.67 | 0.010 | 0.051 |
| OP - PRC | T(53) = -2.24 | 0.029 | 0.098 |
| EC - OP | T(53) = −2.54 | 0.029 | 0.098 |
| Cluster 2 | F(2,52) = 8.16 | 0.001 | 0.003 |
| LOC - IFG tri | T(53) = 4.17 | <0.001 | 0.001 |
| LOC - SFG | T(53) = 3.55 | 0.001 | 0.003 |
| LOC - IFG oper | T(53) = 2.80 | 0.007 | 0.014 |
| IFG tri-LOC | T(53) = 3.21 | 0.002 | 0.019 |
| SFG-LOC | T(53) = 2.65 | 0.002 | 0.019 |
| LOC - HIPP | T(53) = 2.03 | 0.048 | 0.080 |
| IFG tri-EC | T(53) = 2.05 | 0.046 | 0.106 |
| PRC-IFG tri | T(53) = 2.01 | 0.050 | 0.152 |
| Cluster 3 | F(2,52) = 8.14 | 0.001 | 0.003 |
| SFG-HIPP | T(53) = 4.28 | <0.001 | 0.001 |
| IFG oper - HIPP | T(53) = 3.56 | 0.001 | 0.008 |
| HIPP-IFG oper | T(53) = 3.29 | 0.002 | 0.018 |
| IFG tri-HIPP | T(53) = 3.04 | 0.004 | 0.019 |
| HIPP-SFG | T(53) = 2.96 | 0.005 | 0.023 |
| HIPP-IFG tri | T(53) = 2.22 | 0.031 | 0.103 |
| IFG tri-IFG oper | T(53) = 2.17 | 0.035 | 0.106 |

Parametric multivariate statistics (cluster-based inference, functional network connectivity). Cluster-level p-FDR corrected (MVPA omnibus test), connection threshold: p < 0.05 p-uncorrected. PRC perirhinal cortex, EC entorhinal cortex, HIPP hippocampus, LOC lateral occipital cortex, OP occipital pole, SFG superior frontal gyrus, IFG inferior frontal gyrus, tri triangularis, oper opercularis.

**Table 2 | Linear model regression results for the effect of connectivity on MD performance (A)**

| Cluster | Connection | b | SE | p | p-FDR | $R^2_P$ |
|---|---|---|---|---|---|---|
| 1 | LOC-OP | 0.000 | 0.002 | 0.780 | 0.780 | 0.002 |
| | OP-LOC | 0.008 | 0.004 | 0.055 | 0.164 | 0.072 |
| | PRC-OP | −0.003 | 0.002 | 0.130 | 0.260 | 0.045 |
| | OP-EC | −0.002 | 0.002 | 0.469 | 0.586 | 0.011 |
| | OP-PRC | −0.001 | 0.002 | 0.489 | 0.586 | 0.010 |
| | EC-OP | −0.004 | 0.002 | **0.047** | 0.164 | 0.077 |
| 2 | LOC-IFG tri | −0.002 | 0.002 | 0.305 | 0.526 | 0.048 |
| | LOC-SFG | −0.004 | 0.002 | 0.129 | 0.444 | 0.045 |
| | LOC-IFG oper | −0.003 | 0.002 | 0.133 | 0.444 | 0.045 |
| | IFG tri-LOC | −0.001 | 0.002 | 0.533 | 0.609 | 0.008 |
| | SFG-LOC | 0.001 | 0.001 | 0.679 | 0.679 | 0.003 |
| | LOC-HIPP | −0.002 | 0.002 | 0.329 | 0.526 | 0.019 |
| | IFG tri-EC | −0.002 | 0.001 | 0.166 | 0.444 | 0.038 |
| | PRC-IFG tri | 0.002 | 0.003 | 0.513 | 0.609 | 0.009 |
| 3 | SFG-HIPP | −0.005 | 0.002 | **0.010** | **0.017** | 0.126 |
| | IFG oper-HIPP | −0.004 | 0.002 | 0.053 | 0.062 | 0.073 |
| | HIPP-IFG oper | −0.005 | 0.002 | **0.008** | **0.017** | 0.134 |
| | IFG tri-HIPP | −0.004 | 0.002 | **0.019** | **0.026** | 0.106 |
| | HIPP-SFG | −0.008 | 0.002 | **0.002** | **0.013** | 0.177 |
| | HIPP-IFG tri | −0.004 | 0.001 | **0.010** | **0.017** | 0.127 |
| | IFG tri-IFG oper | −0.001 | 0.003 | 0.698 | 0.698 | 0.003 |

A model regressing connectivity on the outcome measure A (age, sex: covariates of no interest) was fitted for each brain connection within the three clusters. Given are the b beta estimates, SE standard error, p values. Within cluster p-FDR correction was applied to the p values. $R^2_P$ partial R square is given to estimate the variance explained by each connection independently of the covariates (age, sex). Values in bold indicate P < 0.05; p-FDR < 0.05 are considered as significant. PRC perirhinal cortex, EC entorhinal cortex, HIPP hippocampus, LOC lateral occipital cortex, OP occipital pole, SFG superior frontal gyrus, IFG inferior frontal gyrus, tri triangularis, oper opercularis.

In sum, lower task-based connectivity during MD between the hippocampus and PFC areas including the SFG, IFG oper and IFG tri is associated with better MD performance, as measured with the A discriminability index.

## Higher connectivity in the hippocampal-PFC network is linked with poorer performance in out-of the scanner memory task measures

In addition to the in-scanner MD task, participants completed several established memory tasks outside the scanner on a separate day. These included the Mnemonic Similarity Task (MST) to assess MD, the Rey–Osterrieth Complex Figure (ROCF) for visual memory and constructional ability, a modified verbal learning and memory test for verbal long-term memory, and the object-in-room recall (ORR) task to evaluate spatial associative memory (see Methods for details).

To examine the link between the LD task connectivity in the hippocampal-PFC network (Cluster 3) and performance in the above independent memory tasks, we conducted a multivariate regression analysis. Given that all the hippocampal-PFC connections were linked to MD performance and showed a similar pattern (Fig. S3), we grouped them together into a composite hippocampal-PFC connectivity score, which was used to simplify the analysis and reduce unnecessary multiple comparisons. This analysis used the LD hippocampal-PFC composite connectivity score as a predictor and a series of memory task measures as dependent variables (Table S2 for details). The results revealed a significant multivariate effect (Pillai = 0.346, F(6, 36) = 3.18, p = 0.013), suggesting that variations in hippocampal-PFC connectivity significantly influence overall performance

on these tasks (Table S1). Specifically, higher connectivity scores were associated with poorer performance on the lure discrimination index (LDI) and the corrected hit rate (Pr) measures of the mnemonic similarity task (MST), which are two alternative metrics to measure MD performance, highlighting a targeted impact (Table S2). Conversely, no significant effects were observed for the other memory measures examined. This pattern underscores the specificity of connectivity effects to particular cognitive processes involved in MD.

### Cognitive-training effects
**Improved mnemonic discrimination performance after training.** The behavioral results from this study were previously reported in Güsten et al. (2024). The 2-week training led to reduced false alarms (incorrect-lures) without affecting hits, suggesting improved performance on correct lures. Here, we further analyzed the bias-corrected A discriminability index. After the baseline scan, where all participants were assessed as one group, they were divided into a training group (n = 26) and an active control group (n = 27).

A two-way mixed ANOVA (covariates: age, sex) testing the Group (training, control) × Time (pre-, post-training) interaction on the A discriminability index revealed a significant effect (F(1,49) = 9.34, p = 0.004, η²G = 0.045) (Fig. 5 and Table S3). Simple effects analyses showed no pre-training difference between the training group (M = 0.842, SE = 0.012) and the control group (M = 0.836, SE = 0.011) [t(49) = 0.361, p = 0.719]. Post-training, the training group (M = 0.917, SE = 0.009) outperformed the control group (M = 0.867, SE = 0.009) [t(49) = 4.018, p < 0.001] (Fig. 5A and Table S3). These results show a training-induced improvement in A discriminability index, driven by increased correct lure responses (Fig. 5).

## Increased intra-occipital cortex connectivity after cognitive training

Next we investigated the effects of cognitive training on the LD contrast. To assess training-induced effects within each of the three clusters identified above (see Fig. 6A–C), we used mixed ANOVAs (Group × Time interaction) with the LD contrast metrics as the dependent variable (covariates: age,

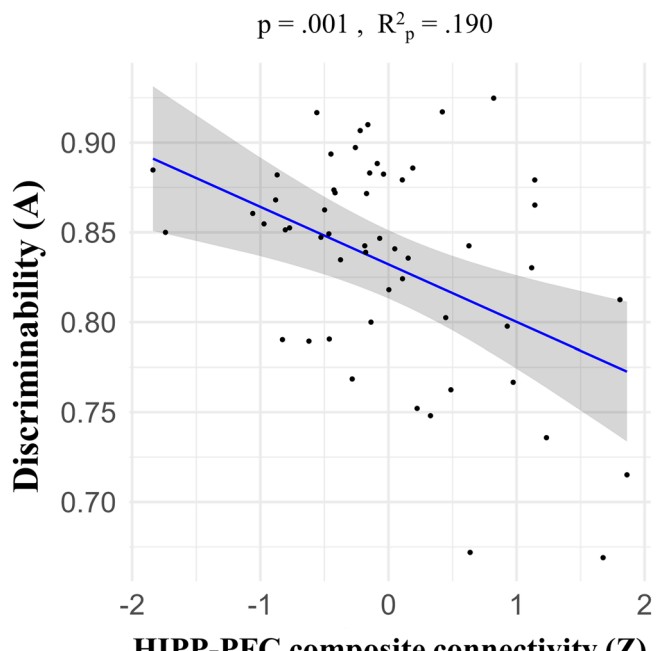

p = .001 , $R^2_p$ = .190

Fig. 4 | Hippocampal-PFC composite connectivity is negatively associated with MD performance. Brain connectivity-behavior linear regression model (*n* = 54) using all the hippocampal-PFC connections from cluster 3 of the LD analysis (Fig. 3 and Tables 1, 2) as a composite z-score. The X axis depicts the independent variable: LD composite PFC-HIPP connectivity values (Z-scores). This composite is the average of the six hippocampal-PFC connections' z-score values. All these connections are part of cluster 3 of the LD analysis (six out of the seven connections in this cluster) and exhibited a significant negative link to MD. For a model fitted for each connection separately, see the supplementary Fig. S3. The Y axis shows the dependent variable: A discriminability. The *p* value and partial *R* square values are shown. In the depicted model, we find a negative link with lower connectivity being associated with higher MD performance.

sex). Via these interactions, we tested whether the connectivity change in each connection was different for the training versus the control group. Since training may have affected only certain connections, we tested a separate model for each connection identified in the baseline LD connectivity analysis (see Fig. 3) to determine whether these were modulated by the training intervention. For detailed statistics for each connection/cluster refer to supplementary Tables S5–S7.

Focusing on the connections of cluster 1, the mixed ANOVAs demonstrated a significant Group × Time interaction effect in the LOC-OP connection ($F(1,49)$ = 10.00, *p*-FDR = 0.016, $\eta^2_G$ = 0.092) (Fig. 6D and Table S4). Post hoc analyses showed that the training group exhibited higher LOC-OP connectivity post- (M = 1.52, SE = 0.994) versus pre-training (M = −5.59, SE = 0.975) ($t(49)$ = −5.12, *p* < 0.001), whereas the control group exhibited no significant difference post- (M = −1.40, SE = 0.951) versus pre-training (M = −2.44, SE = 0.932) ($t(49)$ = 0.78, *p* = 0.439). Conversely, we found no significant training-induced effect in any other functional connection within the cluster 1, 2, or 3 (Tables S5–S7). Therefore, the cognitive-training intervention led to a selective effect of higher task-based LD functional connectivity from the LOC to the OP visual area, but no significant effects for hippocampal-PFC, visual-PFC or any visual to MTL task-based functional connections.

## Post-training hippocampal-PFC connectivity decrease is associated with MD behavioral improvement

Next, to test whether connectivity changes are linked to behavioral improvement, we fitted linear regression models within the training group to investigate the relationship between the post- versus pre-training change in LD connectivity and the A discriminability index (dependent variable) (covariates: age, sex). We focused on the connection LOC-OP (since it exhibited a group-level training effect) and the hippocampal-PFC composite connectivity score (since it was associated with interindividual performance in the baseline data). To ensure that any observed relationship is specific to cognitive training, we also fitted the same model within the control group.

The results revealed that decreases in LD hippocampal-PFC connectivity were strongly associated with greater behavioral improvement in the A index within the training group (*p* = 0.006, *p* FDR = 0.018, $R^2_p$ = 0.291, b = −0.031) (Fig. 7), but not within the control group (*p* = 0.320) (Fig. S5). To ensure this relationship is not disproportionately affected by outliers, we validated the above model using robust regression (robust model: *p* = 0.002) (see Fig. S4). Notably, this relationship was not significant within the control group, as shown using either ordinary linear regression or robust regression (see Fig. S5). For readers interested in the link between each separate

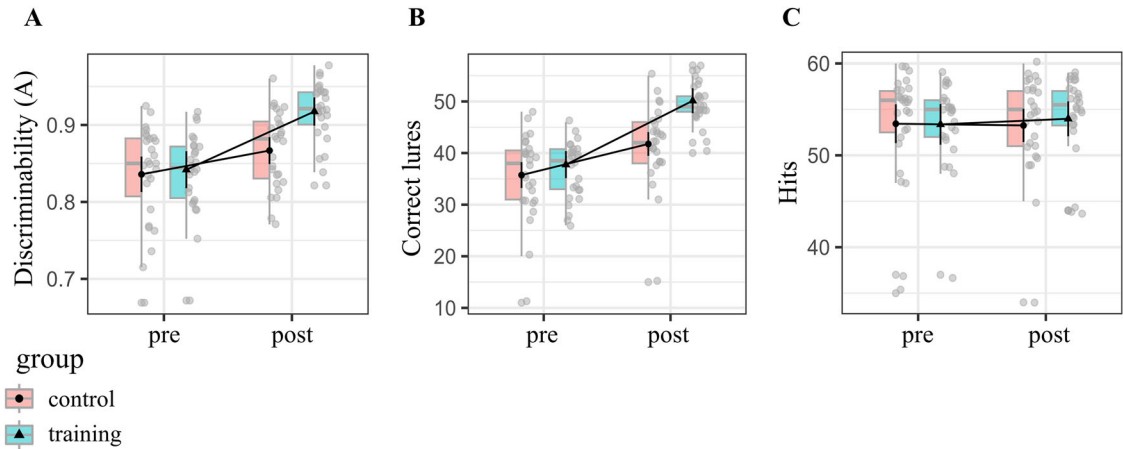

Fig. 5 | Training leads to improved MD behavioral performance. Depicted is the Group x Time (pre-, post-training) interaction shown for different measures: **A** for the discriminability index *A*; **B** lure correct responses; **C** correct repeat trials ("Hits"). We found a significant Group × Time interaction for the *A* Discriminability, which

is driven by the correct lures (see also Güsten et al. 2024). Boxes show the median (center line) and interquartile range (IQR); whiskers extend to 1.5 × IQR; gray points are individual participants. Overlaid black markers/lines are marginal means with 95% confidence intervals (control *n* = 27, training *n* = 26).

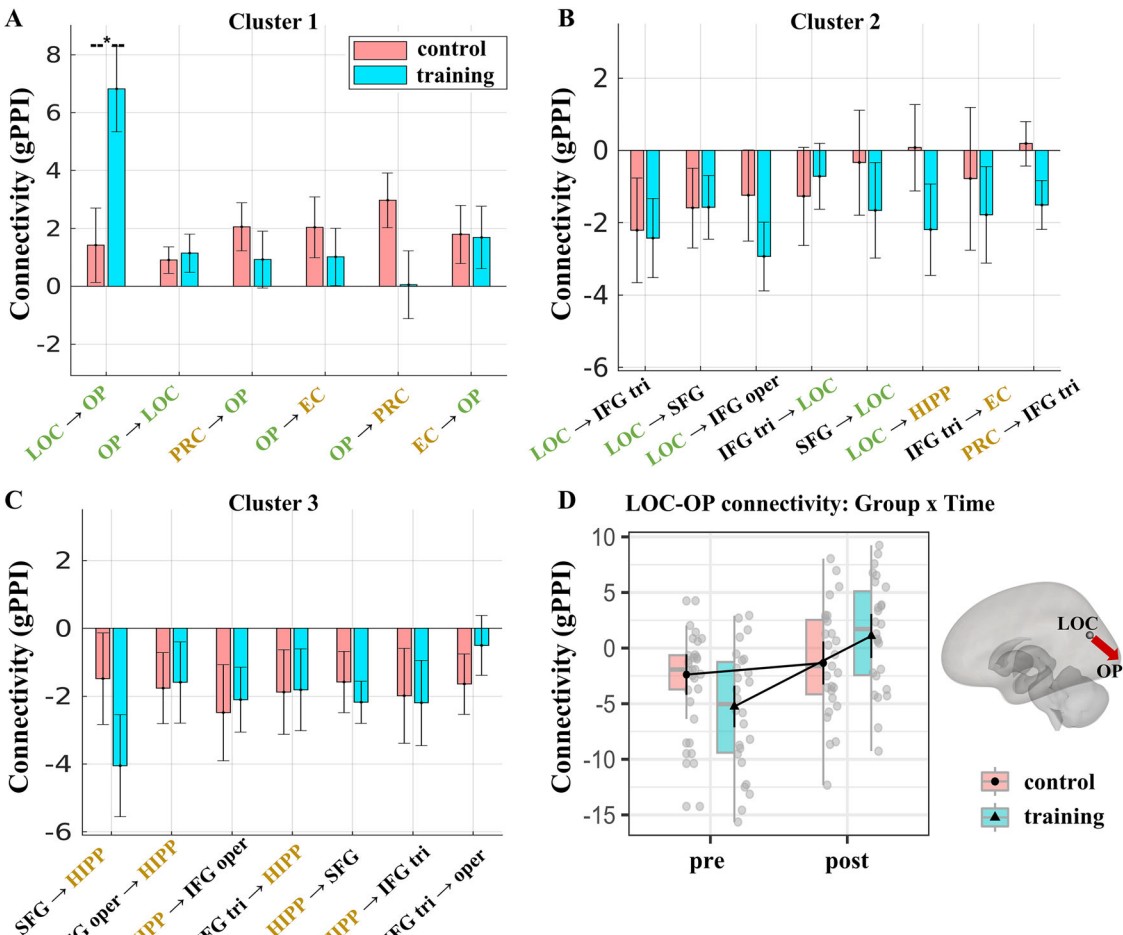

**Fig. 6 | Brain connectivity training effects: increased LOC-OP connectivity after training. A–C** Change in LD connectivity (post-training minus pre-training) in connections of clusters 1–3. Error bars show the standard error of the mean in each group. **D** Interaction plot of the Group x Time (pre-, post-training) effect (ANOVA) for the LOC-OP connection (lateral occipital cortex – occipital pole) ($n = 53$). We found a significant training-induced increase in task-based connectivity from the LOC to the OP. Boxes show the median (center line) and interquartile range (IQR); whiskers extend to 1.5 × IQR; gray points are individual participants. Overlaid black markers/lines are marginal means with 95% confidence intervals (control $n = 27$, training $n = 26$). *$p < 0.05$ p-FDR corrected. *PRC* perirhinal cortex, *EC* entorhinal cortex, *HIPP* hippocampus, *LOC* lateral occipital cortex, *OP* occipital pole, *SFG* superior frontal gyrus, *IFG* inferior frontal gyrus, *tri* triangularis, *oper* opercularis. The color of the ROIs' names denote whether they are a visual (green), MTL (gold), or PFC (black) area.

hippocampal-PFC connection and the MD improvement, this follow-up analysis is provided in the supplementary materials (Fig. S6).

We observed no significant relationship between behavioral improvement and LOC – OP connectivity change ($p = 0.782$, $R^2_p = 0.003$, b = 0.000), or between the hippocampal-PFC connectivity change and behavioral change in the lure discrimination index (LDI) at the MST task ($p = 0.810$, $R^2_p = 0.003$, b = 0.012). Altogether, these results suggest that LD connectivity changes are linked to MD performance improvement, with decreased hippocampal-PFC connectivity being associated with greater improvement. Importantly, the lack of a significant relationship in the control group suggests that this effect is specific to the training intervention.

### Training-related LD changes are lure-driven, and the hippocampal–PFC connectivity-MD improvement association is robust across contrast definitions

To test whether the LD training effect was driven by lure discrimination rather than target recognition (i.e., repeats) and to assess the robustness of our training-related connectivity findings, we conducted two follow-up analyses (see Supplementary sections S2.6, S2.7 for full details). First, a decomposition analysis of the LD contrast showed that the observed LD training effect in the LOC-OP connection was driven by the correct-lures

condition, with no change for repeats (Fig. S8). Second, using a correct-lures versus false alarms (incorrect-lures) contrast, we replicated our primary finding: greater decreases in hippocampal-PFC connectivity were significantly associated with larger MD improvements in the training group ($p = 0.012$, partial $R^2 = 0.252$) but not in the control group ($p = 0.327$) (Fig. S9). These analyses clarify that the LD training effect reflects lure discrimination rather than target recognition, and support the robustness of our main conclusion.

## Discussion

We previously reported that MD training led to improved MD performance and regional functional changes[10]. Here we investigated how functional connectivity patterns of MTL, PFC, and visual brain regions relate to mnemonic discrimination (MD) and inter-individual differences in MD performance before and after cognitive training. We observed a lure detection (LD) functional connectivity signature involving MTL-PFC-visual areas (Fig. 3). Across individuals, higher LD task connectivity between the hippocampus and PFC areas was associated with poorer MD performance, whereby low performers showed higher LD connectivity (Fig. 4). After 2 weeks of MD training, stronger decreases in hippocampal-PFC LD connectivity were associated with greater MD improvement (Fig. 7) selectively within the training group. Finally, training led to greater LD

**Fig. 7 | Hippocampal-PFC connectivity decrease after training is associated with MD improvement in the training group.** (Δ: post- minus pre-train-ing). Linear regression models were fitted within the training group (n = 26). **A** Scatter plot showing the relationship between LOC–OP connectivity chan-ges and MD gains; no significant association was observed. **B** Scatter plot showing the relationship between hippocampal (HIPP)–PFC connectivity changes (composite Z-score) and MD gains; larger decreases in connectivity were associated with larger gains in MD. X axis depicts the independent vari-able: LD connectivity Δ values (gPPI) of a connec-tion (seed ROI – target ROI) or the composite connectivity score. Y axis shows the dependent variable (discriminability index A). Blue lines represent the linear regression fit; shaded bands indicate the 95% confidence interval. P values and partial R square ($R^2_p$) values are shown. $R^2_p$ is given to estimate the variance explained by each connec-tion independently of the covariates. _LOC_ lateral occipital cortex, _OP_ occipital pole.

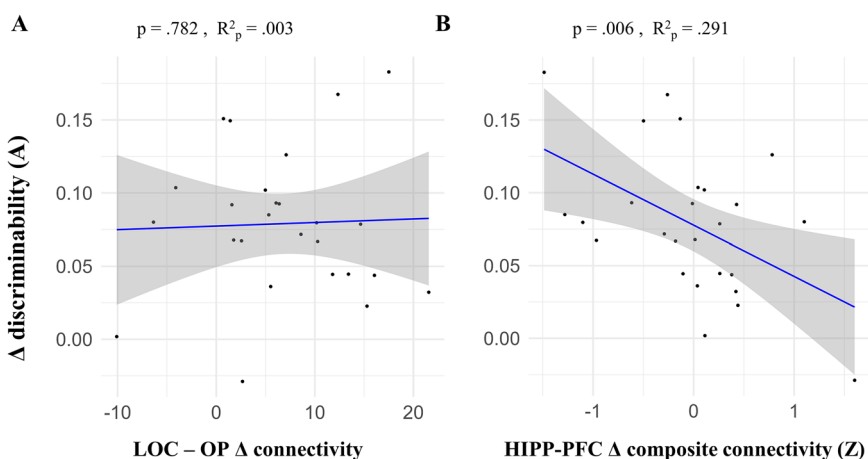

connectivity from the lateral occipital cortex (LOC) to the occipital pole (OP) visual area (Fig. 6).

While the contributions of hippocampal and extra-hippocampal cor-tical regions to MD are recognized[12], their functional interactions during MD remain poorly understood. Our study addressed this gap, demon-strating that task-based functional connectivity patterns within the MTL-PFC-visual network during LD are closely linked to successful MD (Fig. 3). Specifically, we found elevated LD connectivity between the hippocampus and PFC regions (including the inferior frontal gyrus pars triangularis, pars opercularis, and superior frontal gyrus) at the group level.

Our findings align with previous studies suggesting a role for PFC-hippocampus communication in MD[16,17,19,24]. Interestingly, however, higher LD-related connectivity in those hippocampal-PFC connections predicted poorer MD performance within our sample (Fig. 4). One explanation is that increased hippocampal–PFC connectivity may reflect neural inefficiency[25]. Individuals with poorer MD performance might over-recruit PFC regions, in synchrony with the hippocampus, without experiencing a benefit for task performance. Our findings raise the possibility that over-connectivity with PFC regions could interfere with the hippocampus's specialized role in fine-grained discrimination required for MD. This interference may be due to an introduction of neural noise (animal findings show that prefrontal projec-tions modulate hippocampal signal-to-noise ratio[26]), a failure to suppress irrelevant information[27] or due to competition between PFC and hippocampus-based memory processing. Competition may be due to involvement of PFC working memory mechanisms that conflict with hip-pocampal MD processing[28–30].

Instances of over-recruitment without performance benefits have been reported in earlier studies[31]. One framework explaining such a negative relationship is the balance between functional integration and segregation within neural networks. According to this framework, optimal cognitive performance depends on a dynamic balance: integration enables commu-nication across distributed regions, while segregation supports specialized processing within specific circuits. Excessive integration—manifested as increased connectivity—can disrupt specialized processing and lead to decreased performance on tasks relying on specific neural circuits[32]. Several human neuroimaging studies show that hyperconnectivity can be linked to impaired cognitive function and reduced network efficiency, a pattern also observed in several pathological brain states[33,34]. For example, hippocampal hyperconnectivity has been associated with poorer MD performance in cognitively normal older adults[35] as well as worse memory performance in patients with mild cognitive impairment[36]. Extending these findings, our results in young adults and specifically using MD-based task-connectivity

data suggest that hyperconnectivity can be a marker of poorer memory performance even within healthy young populations, indicating that the integration-segregation principle may apply across the adult lifespan.

In the case of MD, heightened hippocampal–PFC connectivity might reflect an imbalance where excessive integration interferes with hippo-campal function. A failure to maintain optimal segregation may allow non-specific or interfering influences to disrupt the processing required for successful discrimination. This interpretation aligns with the Inhibitory Deficit Theory, which suggests that cognitive performance suffers when there is insufficient ability to filter out irrelevant or interfering information[37]. Network-level studies further support that optimal cognitive performance depends on dynamic shifts in brain organization, where both excessive integration and poor segregation can impair function[32,38,39]. Thus, more connectivity is not always better, and optimal MD may require a fine-tuned balance between hippocampal specialization and its integration with pre-frontal networks. Our results motivate future causal research in animals and brain stimulation where hippocampal-PFC connectivity can be directly manipulated to study these mechanisms.

Importantly, this relationship between higher LD hippocampal-PFC connectivity and lower MD performance was observed both during the fMRI paradigm and in an independent MD task performed out-of-scanner: the mnemonic similarity task - MST[18] (Tables S1, S2). Notably, hippocampal–PFC connectivity did not significantly relate to performance on other out-of-scanner memory tasks, such as the verbal learning, complex figure, or object-in-room recall tasks, which rely on broader memory or cognitive processes. This association was evident for both metrics of MD performance extracted from the MST task—the lure discrimination index (LDI) and the corrected hit rate—supporting the robustness of the finding. This specificity supports the interpretation that hippocampal–PFC con-nectivity is particularly relevant for the fine-grained discrimination pro-cesses required in MD, rather than reflecting a general association with broader memory or cognitive abilities. Altogether, these suggest that hippocampal-PFC connectivity during MD-related processing may be a robust task-invariant individual difference marker of MD performance. To our knowledge this is the first time that inter-individual differences in brain function (task-functional connectivity) related to performance across two different MD tasks have been identified. Further causal research is war-ranted to elucidate the functional role of hippocampal-PFC connectivity in MD, potentially through manipulations that modulate prefrontal or hip-pocampal activity, such as transcranial magnetic stimulation or focussed ultrasound stimulation[40]. Future studies could also examine how the task-based connectivity patterns identified here relate to MTL-associated

functional connectivity networks observed during resting-state, to further elucidate the generalizability and specificity of these findings.

Given the limitations of fMRI in capturing the temporal neural dynamics of functional connectivity, more direct measures, such as electrophysiological recordings of neural oscillations, could provide valuable insights into the mechanisms underlying hippocampal-PFC communication in MD. Theta oscillations have been proposed as a potential mechanism orchestrating memory-related hippocampal-prefrontal interactions[20,41–44]. Further electrophysiological research is needed to determine the extent to which hippocampal-PFC coupling via theta oscillations supports MD and to elucidate the directionality of this communication. Additionally, brain stimulation techniques targeting PFC theta oscillations could test the causal role of theta-mediated connectivity in MD and explore its potential to enhance cognitive performance by modulating hippocampal-PFC communication.

In addition to hippocampal-PFC interactions, our findings highlight the role of visual cortex connectivity during MD. Our baseline pre-training results suggest that successful MD relates to a wider cortical network encompassing both lower and higher-order visual areas, interconnected with prefrontal and extra-hippocampal MTL regions (Fig. 3 and Table 1). We interpret that these connectivity patterns support perceptually fine-grained, high-fidelity representations necessary for discriminating similar memories[17]. Our findings extend previous studies showing the involvement of occipital visual areas in MD[13,15] with new data on their functional connectivity. This inter-cortical communication may facilitate cortical pattern separation potentially by fine-tuning visual input to the hippocampus and regulating the influence of top-down prefrontal control[12,13,45]. This interpretation is consistent with research highlighting the role of top-down influences on visual processing, including the modulation of early visual areas by higher-order regions[45].

Our 2-week cognitive training intervention had an impact on visual connectivity. We observed increased connectivity from the LOC to the OP after training, suggesting training-induced plasticity in a network critical for visual processing (Fig. 6 and Table S4). This finding demonstrates that our behavioral intervention modulated neural circuits within the visual system and extends previous research suggesting plasticity of visual cortical areas following training[46,47], including functional reorganization in the early visual cortex[47]. However, the precise functional role of this change for MD improvement remains to be disambiguated, as we did not find a significant correlation between the magnitude of this LOC-OP connectivity change and individual gains in lure discrimination. This presents an interesting dissociation: while our intervention induced plastic changes in visual networks, only the changes in prefrontal-hippocampal connectivity were linked to individual differences in MD gains (Fig. 7). Future research is needed to clarify this relationship and test specific mechanisms, for example, whether enhanced visual representations facilitate downstream mnemonic processes by providing the hippocampus with more distinct inputs.

Our results show that while the training intervention did not produce a significant change in hippocampal-PFC connectivity at the group level, the degree of individual change was meaningful. Individuals in the training group who showed a greater post-training decrease in hippocampal-PFC connectivity also exhibited larger improvements in MD performance (Fig. 7). This relationship was absent in the control group (Fig. S5), confirming that this brain-behavior link is specific to those who underwent the intervention. This conclusion is supported by our sensitivity analyses demonstrating that this brain-behavior relationship holds true also when using a correct- vs incorrect-lures fMRI contrast, confirming its robustness (see also Supplementary section S2.7). Thus, the connectivity pattern that explained baseline performance variability, also explained inter-individual differences in performance improvements after training. This reinforces the idea that hippocampal-PFC connectivity is a neural resource for MD performance that not only explains individual differences but can also be targeted to improve performance. This pattern—a significant brain-behavior correlation in the intervention group, in the absence of a group-level main effect—is common in intervention studies and suggests that individuals

respond to training heterogeneously[48–53]. Training may have promoted a more efficient division of labor between PFC and hippocampus, where the hippocampus performed pattern separation more effectively with less reliance on top-down control from the PFC. Studies have shown that training can decrease PFC engagement, indicating increased processing efficiency[54].

An important implication of our findings relates to the specificity of this training effect. The baseline association between hippocampal-PFC connectivity and MD generalized to an independent task (the MST). In contrast, the post-training change in connectivity was only associated with performance improvements on the trained task, not the MST. This suggests our intervention induced a task-specific "near transfer" effect, rather than a more generalized "far transfer" to another MD task. The lack of a group-level effect on connectivity, combined with the task-specific nature of the brain-behavior correlation, suggests that our two-week intervention was sufficient to modify the specific neural circuits engaged by the task, but perhaps not long or varied enough to induce a more global reorganization of the MD network that would generalize to untrained tasks.

While our 2-week training protocol revealed significant changes in visual cortex connectivity, longer interventions may be necessary to induce more widespread network changes. Future research should investigate whether extended training leads to broader functional reorganization, including changes in hippocampal-PFC connectivity, and whether these changes are associated with MD improvement transfer to non-trained MD tasks and other episodic memory tasks. Our study focused on young adults and so paves the way for future studies to investigate the efficacy and neural correlates of similar training in older adults, particularly those experiencing cognitive decline. Understanding how these training programs affect brain function in the context of aging and neurodegeneration is crucial for developing effective interventions to maintain cognitive health in older adults.

By using a memory training intervention, our study provides compelling evidence for the role of brain connectivity in MD. To strengthen causality, future research utilizing brain stimulation techniques can be helpful. For instance, applying inhibitory stimulation to the PFC during MD tasks could help determine whether disrupting PFC activity leads to improved performance in individuals who initially exhibit high hippocampal-PFC connectivity.

In conclusion, we found a network of functionally connected MTL-PFC-visual areas involved in MD and identified a hippocampal-prefrontal brain connectivity signature of MD that generalizes across two independent tasks. This hippocampal-prefrontal neural signature also explained individual performance gains after training. Thus, we identified a neural resource for MD performance that not only explains individual differences but can also be targeted to improve memory performance.

## Materials and methods
### Participants
Sixty young adults were recruited at the Otto-von Guericke University of Magdeburg. Five participants voluntarily chose to discontinue with the study and one was excluded due to problematic brain extraction (M age = 23.76 years, SD = 3.35, 61.11% female). One further subject was excluded from the connectivity training analysis due to incomplete fMRI data, resulting in a total sample of 54 for our baseline analyses and 53 for our 2 × 2 training analyses. All participants were screened for prior psychiatric or neurological diagnoses and had intact or corrected vision. All participants provided written informed consent and were compensated for their time. All ethical regulations relevant to human research participants were followed. The study was approved by the Otto-von Guericke University Magdeburg ethics committee and conducted in accordance with the declaration of Helsinki, 2013.

### Study design, procedure, and experimental paradigm
We used a 2 × 2 longitudinal intervention design (Group × Time). Participants were randomly assigned to an experimental training group (n = 26) or an active control group (n = 27) with a task fMRI session

**Fig. 8 | Training design overview.** Participants underwent a computerized training, from a remote location (mobile). Both training and control groups underwent active training, but on different tasks. **A** The training group (*n* = 26) performed a six-back version of the object-scene task[4,8], with separate object and scene trials, and domain-specific level increase. Level difficulty was increased by showing more difficult images based on previously evaluated difficulty. Stimulus discriminability rather than trial length was modified to more directly target discrimination ability. **B** The control group training (*n* = 27) consisted of eliminating moving neurons by clicking on them. With level increase, neurons became more numerous and faster. To keep stimuli visually similar to the main task, background images were equivalent to the scenes from the training task. Figure modified from Güsten et al (2024). For more details on the training see Güsten et al. (2024).

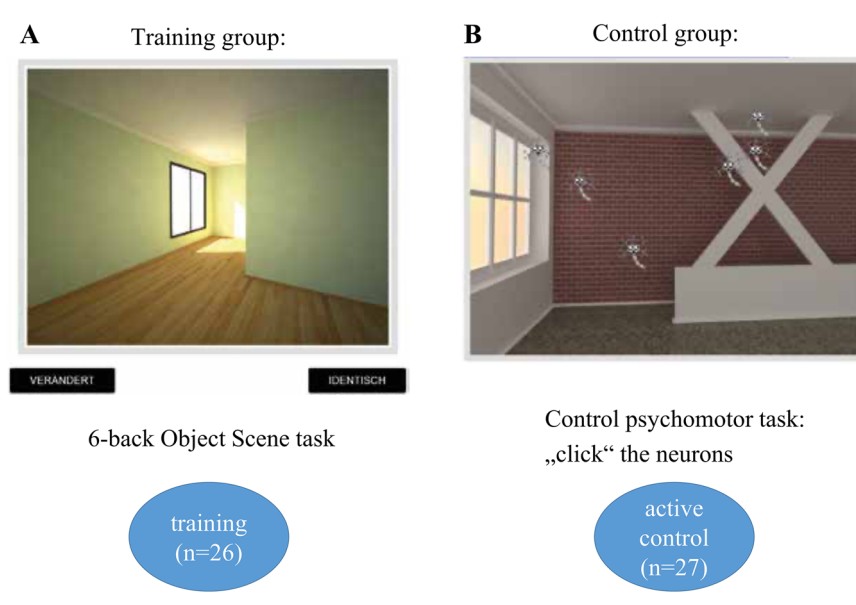

completed pre- and post- a 2-week computerized web-based cognitive training intervention (Figs. 1, 8). Pre- and post-training, participants performed at the scanner, while fMRI data was collected, a six-back version of the object-scene task developed to measure MD[4,8] (see Fig. 1D). Before scanning, subjects were given standardized instructions and underwent a short training session. Stimuli were presented via a mirror on a magnetic resonance (MR) compatible display. When necessary, vision was corrected using MR-compatible glasses.

The object-scene task is a continuous old/new recognition task (Fig. 1A–C), described in detail in Berron et al. (2018). In this task, subjects have to recognize whether each item presented is a "new" or an "old" (repeated) image by responding with their right index or middle finger, while no indication is given to the participants that the encoding block has ended. We adapted it for our study by using a 6-back instead of the original 2-back format, to boost difficulty given participants' young age and expected post-training task proficiency. The task had 2 × 13 min runs. Stimuli were presented in sequences of 12 items each (six-back format: six items presented in encoding, six in test phase). A sequence consisted of either object or scene stimuli only presented on a gray background. In each sequence, the first 6 stimuli were shown for the first time ("first" trials; correct response: "new"), while each of the following six stimuli (test phase) could be either an exact repetition ("repeat" trial; correct response: "old") or a very similar yet different version of the first images ("lure" trial; correct response: "new"). In lure trials, local features of the objects or global features (e.g., the geometry) of the scenes were changed (Fig. 1C), but not the color, position, size or viewpoint. Ten gray noise images ("scrambled") were presented as perceptual control trials at the start and end of each run (40 trials in total). In total, 60 sequences were shown. This resulted in a 2 × 2 factorial design with 30 trials for each trial type combination (lure objects, repeats objects, lure scenes, repeats scenes). The sequences' order was counterbalanced with respect to the object–scene, and repeat–lure trial types. The stimuli were presented in an event-related design with each stimulus shown for 3 s, being separated by a fixation star. Jittered inter-stimulus intervals were used (range: 0.8 to 4.2 s, mean ~1.63 s) to optimize statistical efficiency[55]. The task difficulty (correct response rate) for each image was validated in an independent study[4], ensuring that different stimuli sets of similar difficulty were used at the two fMRI sessions. Note that the stimuli used for the cognitive training were created using the same principles.

It is important to note that although previous univariate analyses of these data have examined potential differences between object and scene stimulus types, the present study was not designed to test category-specific

effects, and our hypotheses did not pertain to stimulus modality. Importantly, the cognitive training intervention included both object and scene stimuli, and prior analyses[10] found no significant interaction between stimulus type and training effects. Therefore, we collapsed across stimulus types in all analyses.

## Cognitive training

In between fMRI scans, all participants performed a 2-week computerized cognitive training on an online platform (http://iknd-games. ovgu.de/MemTrain/), comprising six training sessions in total (Fig. 8). The training group performed the 6-back object-scene task described above. The stimuli were chosen from items assessed in Güsten et al. (2021). The training consisted of 45 different levels in total. Subjects progressed to the next level upon reaching a threshold of correct responses (ranging from 70% up to 90% across the levels), otherwise they would repeat a level until they achieved the threshold. At each level, the set of stimulus pairs was fixed, and the presentation of each stimulus version (1 or 2) was randomized for both the presentation and test trial. The levels were created with increasing difficulty as one progressed in the training. The task consisted of rounds. Each round was made up of four blocks (two object and two scene blocks). Each block contained a presentation and test phase. First, six stimuli were presented for 3 s each, separated by a blank page (1.5 s). Then, a countdown (3 s) indicated the test trials, in which participants had to indicate whether a presented stimulus was a repeat or a lure (max. 4.5 s). A training session ended after spending 45 min in the training rounds.

The control group trained on a psychomotor task in which they had to click on small moving neuron-icons presented on top of background images (see Fig. 8). To keep the visual stimulation similar to the actual training task, the background images were pseudorandomly taken from the stimulus set of the training group. The training structure was the same as in the main training task to keep the training flow comparable. Namely, one training round consisted of four blocks: two blocks with objects and two blocks with scene stimuli appearing in the background. A higher level was reached by eliminating a certain percentage (varied from 80 to 90% across the levels) of neurons within a round. Again, a training session ended after spending 45 min in the training rounds. The control task was progressively made more challenging at higher levels in several ways: (1) reducing the neurons' presentation time, (2) increasing the number of neurons to eliminate, (3) increasing neurons' moving speed, (4) making neurons change direction, and (5) introducing "bad" neurons, which participants had to avoid clicking.

## Out-of-the scanner cognitive tasks

In addition to the experimental paradigm performed in the scanner, all subjects completed a series of transfer tasks as explained in detail in ref. 10. Here we examined whether task-related connectivity in our MD task, which was performed at the scanner, is associated with a series of independent memory tasks performed outside of the scanner at another day (part of the transfer tasks) (see Table S2). Those memory tasks included the mnemonic similarity task (MST)[18,56], a modified 30-word list verbal learning memory task[57], the Rey–Osterrieth complex figure (ROCF)[58], and the object-in-room recall (ORR) task[59].

The MST (Version 0.9)[56] assessed mnemonic discrimination through separate encoding and retrieval phases. In the encoding phase, participants made indoor-outdoor judgments for 128 object stimuli. The retrieval phase involved identifying 192 stimuli (64 repeats, 64 similar lures, and 64 foils) as either repeats, lures, or foils. The LDI and corrected hit rate measures were calculated. The LDI measures the ability to detect lures while controlling for response bias. The corrected hit rate estimate (Pr) ("old"|old - "old"|lure) is an alternative bias-corrected measure of mnemonic discrimination performance, which is calculated based on the hits (correct repeats) minus the false alarms (incorrect-lures)[60]. To assess memory retrieval and pattern completion, the ORR task was employed. The ORR task (a digital version of the task is reported in ref. 59) involved participants memorizing rooms containing objects and their specific locations. It consisted of an encoding phase, where participants memorized 75 room-object combinations, followed by an immediate retrieval test to ensure accurate encoding. During retrieval, participants viewed a room with a cued location and selected the correct object from three options: the correct object, an object previously present in the picture but in a different location (internal lure), and an object not present in the room at all (external lure).

To assess visual memory, visuospatial constructional ability, and recognition, the Rey–Osterrieth complex figure (ROCF) was used. Participants first copied the figure, then reproduced it from memory after 3 min, and again ~30 min later. For verbal long-term memory and interference, a modified version of the verbal learning and memory test (VLMT) was applied, the lists were extended to a length of 30 words by adding the lists from the CVLT. Participants were exposed to lists A and C, with interference introduced through the VLMT's 15-word list, followed by list D in the posttest. The procedure involved the experimenter reading the word list five times, each followed by immediate recall (L 1–5), then an interference list recall, a recall of the original list (L 6), and a final recall (L 7) after a 30-minute delay. For a more detailed description of the tasks and associated measures, see also ref. 10.

## Imaging data acquisition

All imaging data were acquired on a 3T MAGNETOM Skyra scanner (Siemens, Erlangen, Germany) using syngo MR E11 software and a 64-channel head coil. Structural images were acquired using a T1-weighted MPRAGE sequence with 1 mm isotropic resolution (3D acquisition; TR/TE/TI = 2500/4.37/1100 ms, FoV = 256 × 256 mm²; flip angle = 7 degrees; BW = 140 Hz/Px). Whole-brain functional data were acquired in two runs of 13 min each, using T2*-weighted echo planar imaging (EPI). We used a 2D simultaneous multi-slice EPI sequence (SMS-EPI) developed at the Center for Magnetic Resonance Research, University of Minnesota (CMRR; Moeller et al., 2010), with a 2 × 2 mm² resolution, FOV = 212 × 212 mm², TR/TE = 2200/30 ms, 10% slice gap, multiband acceleration factor 2, GRAPPA 2, phase encoding (PE) direction P > A, 64 slices and 2 mm slice thickness. In addition, phase maps were acquired with the following parameters: TR/TE1/TE2 = 675/4.92/7.38 ms, spatial resolution = 3 mm, and 48 slices.

## MRI preprocessing and denoising

We processed MRI data using the "fMRIPrep" (v. 20.2.6) pipeline[61], employing default options, and the CONN toolbox[62]. Each T1w image was corrected for intensity non-uniformity and skull-stripped. A T1w-reference map was computed by registering 2 T1w images acquired from each subject (which were collected within a 2-week period) to obtain a robust estimate of anatomy. Spatial normalization to the MNI space (ICBM 152 Nonlinear Asymmetrical template 2009c) was performed through nonlinear registration. Tissue segmentation was performed on the brain-extracted T1w images. The functional data were slice-time corrected and co-registered to the T1w using boundary-based registration with 9 degrees of freedom. Physiological noise regressors were extracted using CompCor, in addition to other confound variables, including head-motion parameters and framewise displacement (FD), which were used later at denoising. Frames that exceeded a threshold 0.5 mm FD were annotated as motion outliers. All resampling was performed in a single interpolation step. For further details, including the software used in each step, refer to the documentation: https://fmriprep.readthedocs.io/en/20.2.6/.

Next, the functional data were denoised using the CONN toolbox to remove potential confounding effects in the BOLD signal. The denoising steps included: regressing out three translation and three rotation motion parameters plus their first-order derivatives (12 parameters in total), anatomical component-based noise correction procedure (aCompCor) to remove noise components from the white matter and cerebrospinal areas, scrubbing to remove the volumes identified as outliers due to excessive motion (>0.5 mm FD), high-pass filtering 1/128 Hz, linear detrending, and despiking. Covariates were also included to account for potential slow trends, initial magnetization transients, or constant task-induced responses in the BOLD signal, according to the default denoising pipeline in CONN[63]. For further details, see the documentation: https://web.conn-toolbox.org/fmri-methods/denoising-pipeline.

## Functional connectivity analyses

**Regions of interest (ROIs).** We performed hypothesis-driven region-of-interest (ROI)-to-ROI analysis. For each ROI, unsmoothed BOLD data were extracted by default in CONN. We constructed a model including major medial temporal lobe (MTL), prefrontal (PFC), and visual ROIs: hippocampus, entorhinal cortex (EC), perirhinal cortex (PRC), parahippocampal cortex (PaHC), superior frontal gyrus (SFG), middle frontal gyrus (MidFG), and inferior frontal gyrus (IFG) including pars triangularis (tri) and pars opercularis (par), medial prefrontal cortex (medFC), lateral occipital cortex (LOC) and occipital pole (OP) (Fig. 3). All ROIs were bilateral at MNI152 space. The EC was extracted from the Juglich atlas[64], the PRC and PaHC from established MNI-space segmentations[65] (https://neurovault.org/collections/3731/). The remaining ROIs were derived from the Harvard-Oxford cortical and subcortical atlases[66].

**Task-modulated connectivity metric (gPPI) and MD contrast.** We applied a region-of-interest generalized psychophysiological interaction (gPPI) analysis[23] to examine the task-modulated functional connectivity between the ROIs in our model (ROI-to-ROI analyses) for the successful MD contrast (*lure detection - LD*: correct lure minus repeat trials). All functional connectivity analyses performed in our study refer to this LD contrast. We used the CONN toolbox[63] for all analyses. gPPI values were calculated for each ROI-to-ROI connection pair for each participant, representing LD task-functional connectivity from a seed to a target region. gPPI measures the influence of a seed on a target region after partialling out task-related activity and task-unrelated connectivity, resulting in a functional connectivity matrix. This means that each connection given follows a seed–target format.

**Characterization of baseline LD profile.** To typify the connectivity patterns associated with successful MD, we conducted analyses of the LD contrast in the pre-training whole-sample data, without respect to subsequent group allocation. This resulted in an 11 × 11 gPPI matrix (11 ROIs, 110 connections). We used a cluster-level inference with the default ROI-to-ROI network multivariate parametric statistics approach in CONN. This approach uses multivariate statistics to examine groups of related connections, resulting in an F-statistic for each cluster with FDR-corrected cluster-level *p*-values, in addition to a post-hoc connection-

level thresholding that keeps the strongest connections within each significant cluster. We used the standard settings: $p < 0.05$ cluster-level $p$-FDR, connection threshold: $p < 0.05$ $p$-uncorrected. Thus, connections were considered related if they formed part of a cluster that survived FDR correction at the cluster level, and only those connections within significant clusters that also met the connection-level threshold ($p < 0.05$ uncorrected) were retained for further analysis. This two-step thresholding approach ensured robust, statistically meaningful patterns of connectivity. For further details on the method, see the CONN toolbox documentation[67].

**Cognitive-training effects on LD connectivity.** To test the training effects on the ROI connectivity data, we performed mixed $2 \times 2$ analysis of variance (ANOVA) (aov_ez default type III sum of squares option, R package "afex" v. 1.3-0)[68]. As the dependent variable, we used the gPPI connectivity values from each connection in each of the clusters identified from the previous baseline characterization step. We added the Group (training, control: between-subjects factor) × Time (pre-, post-training: within-subjects factor) as independent variables, the sex and age as covariates. We focused on the Group × Time interaction effect to probe the training effects. We applied Hyunh-Feldt correction for sphericity violations cases, and calculated both the generalized Eta-squared ($\eta^2_G$) and partial Eta-squared ($\eta^2_p$) to estimate the effect size for the ANOVAs[69]. Post hoc tests were conducted with the emmeans R package v. 1.8.8[70]. When required, simple effects and an interaction contrast were calculated to test if the training-induced (post- versus pre-training) change was bigger for the training compared to the control group. We applied p-FDR multiple comparisons correction for the tests performed within each cluster of connections to strike a statistical power and control balance.

**Behavioral data processing and cognitive-training effects**
The behavioral data from the MD fMRI task were processed using R (v. 4.3.1)[71]. We calculated four readouts; hit rates (repeats correct), correct rejection rates (lures correct), false alarm rates (lures incorrect), and the $A$ discriminability index (a non-parametric discriminability estimate based on the hits minus false alarms rate and uses a corrected formula to estimate the non-parametric sensitivity associated with the signal and noise distributions)[72]. This $A$ index is regarded as an improved alternative to A prime. We calculated total scores by combining both modalities (objects, scenes), as we were interested in the total modality-independent MD performance.

To test the training effect on MD performance, we employed $2 \times 2$ mixed ANOVAs (Factors: Group × Time, covariates: sex, age) following the same statistical approach as for the ROI connectivity data analyses. We used the A discriminability Index as the main dependent variable, but also examined the underlying trial types separately: correct lures and hits (correct repeats).

**LD connectivity–behavior regression**
To examine the LD connectivity-behavior relationship, we fitted linear regression models in the pre-training data ("lm" R function). Each model used the formula: A ~ connectivity + age + sex (A discriminability: dependent variable. Connectivity: values of a certain connection as the independent variable). Age and sex were included as covariates of no interest. We applied $p$-FDR multiple comparisons correction for the tests performed within each cluster of connections. To estimate the variance explained by a connection, we calculated the partial $R$ square ($R^2_p$) using the "sensemark" R package v. 0.1.4[73].

**Relationship between LD hippocampal-PFC connectivity and other memory tasks**
To test the association between LD task connectivity in the hippocampal-PFC network and performance on various out-of-the-scanner independent

memory tasks, we conducted a multivariate regression analysis using the LD hippocampal-PFC composite connectivity score as the main predictor. This score was calculated as the average of the z-score transformed connectivity values extracted from each of the six hippocampal-PFC connections values (which belong to cluster 3 and were also found to be linked to high performance). Dependent variables included performance scores from multiple memory tasks including the MST task (measures: LDI and corrected hit rate), the ORR task (measures: correct retrieval performance, and retrieval internal errors), the Rey–Osterrieth Complex Figure (measure: delayed recall), and VLMT test (word list test: encoding vs delay recall score (List5 minus List7 performance)). Age and sex were included as covariates. The Pillai's statistic was calculated to assess the overall model fit and determine the significance of the connectivity score effect across the different memory measures using the "Anova" function ("car' package) and the "lm" function in R. Then we examined using linear models the relationship of the hippocampal-PFC composite connectivity score with each separate memory measure (dependent variable).

**LD connectivity change–behavior change regression**
To examine the LD connectivity-behavior change ($\Delta$) relationship (post-minus pre-training), we fitted linear regression models ("lm" R function) within the training group; we focused on the LOC-OP connectivity (for which we observed a group training effect), and hippocampal-PFC composite connectivity score (which was associated with MD performance at the baseline data). Each model used the "A Discriminability $\Delta$" as the dependent variable, the "connectivity $\Delta$" as the independent variable, while "age" and "sex" were included as covariates of no interest. To estimate the variance explained by the connectivity change, we calculated the partial $R$ square. To ensure that outliers did not disproportionately affect the hippocampal-HPC connectivity-behavior change results, we further validated our model using robust linear regression that is less sensitive to outliers (Fig. S4) (MM-type estimator; "lmrob" function from the "robustbase" R package, version 0.99-4-1)[74].

**Statistics and reproducibility**
In all statistical tests, we used $\alpha = 0.05$. Where applicable, we applied FDR correction and provided both the raw $p$ values and the $p$-FDR corrected values. Beyond $p$ values we provide effect size measures such as the partial $R$ square (linear regression models), as well as both the generalized and the partial HTA square (ANOVAs) so that the reader can evaluate both $p$ values and effect size of the reported effects. Furthermore, we replicated our main connectivity-change behavior-change relationship using an alternative fMRI contrast, which further supports our conclusions. Our sample consisted of 54 subjects; for the connectivity analyses, 53 subjects were included. Our design also employed a control group, which strengthens our study's methodological rigorness. The baseline pre-training task connectivity analysis, which identified the different clusters of connectivity, was conducted using the CONN toolbox. In CONN, we applied established cluster-level and multiple comparisons correction methods (see the Methods section for details). The targeted ROI-based connectivity group comparisons and connectivity-behavior regression models were conducted in the R statistical software, where we performed the statistical testing and $p$-FDR corrections. Details on the steps and statistical packages we used are provided in more detail in each of the Method sections, respectively.

**Reporting summary**
Further information on research design is available in the Nature Portfolio Reporting Summary linked to this article.

**Data availability**
The data employed in this study are not publicly available. The source data for the figures and tables are provided as Supplementary Data 1, 2, respectively.

## Code availability
No custom code or algorithms were used in this study. All analyses were performed using publicly available software tools as described in detail in the Methods section.

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

## Acknowledgements

This work was supported by Deutsche Forschungsgemeinschaft (DFG, German Research Foundation)—Project-ID 425899996—SFB 1436. We are grateful to Aditya Nemali for his help in setting up the training website. We thank all participants who volunteered to participate in the training. We also want to thank the staff at the IKND and the university clinic for neurology, Medical Faculty, Otto-von-Guericke University, for assistance in testing and scanning the participants.

## Author contributions

Conceptualization: P.I. and E.D. Methodology: J.G., E.D., and P.I. Investigation: J.G. (performed the experiment). Data curation: J.G. Formal analysis: P.I. Writing—original draft: P.I. Writing—review & editing: P.I., E.D., E.M., J.G., A.M., R.C., and F.K. Supervision: E.D. and A.M. Funding acquisition: E.D. Project administration: P.I. (managed manuscript revisions and correspondence during peer review).

## Funding

## Competing interests

The authors declare no competing interests.
