## [Transparent Peer Review file · Communications Biology]

Hippocampal-prefrontal connectivity relates to inter-individual differences and training gains in distinguishing similar memories

Corresponding Author: Mr Panagiotis Iliopoulos

Version 0:

Reviewer comments:

Reviewer #1

(Remarks to the Author)

The authors investigated fMRI-based functional connectivity and its relation to interindividual differences in memory discrimination (MD), and whether connectivity changes would be associated with MD improvements after cognitive training. Results showed a negative brain-behavior relationship: decreased hippocampal-prefrontal connectivity was linked to better MD performance across individuals and to larger improvements after cognitive training. Additionally, cognitive training increased functional connectivity between higher visual areas. The results are interesting, but major arguments in the introduction and discussion could be strengthened. I also have some concerns about their analytical approach. Overall, I hope these comments are helpful in improving the manuscript.

Major points:

I struggle to fully grasp the author's goal here: "An overarching question is whether neural resources explaining interindividual variability also explain training-related gains. If we find convergence (...) it would indicate that high performers and beneficiaries of cognitive training tap into similar neural resources". This appears trivial to me, but maybe I am missing the point. Why would you even assume a divergence between these aspects? The authors go on to explain what convergence / divergence would mean, but I struggle to see the point. Are there other examples from the literature where a potential divergence between interindividual variability and training-gains was reported?

As long as the authors are not performing DCM or a variant thereof, I urge them to refrain from any comments on effective connectivity or directionality. gPPI is an extension of PPI, but it is essentially still a regression model, so, even if one defines a "seed" and a "target", findings should be interpreted as increased/decreased functional connectivity between x and y, but not as effective connectivity from x to y. Similarly, while the precise connectivity profile clearly depends on the seed, the gPPI analysis will likely yield overlapping results when taking x or y as a seed. This is also visible in their results (e.g., Table 2), where many of the connections that are significantly linked to LD appear as two sides of the same coin. Considering this may help remove some of the redundancy and make the results more on point.

Head motion can substantially impact functional connectivity results and induce spurious findings. Did the framewise displacement differ between groups and between pre- vs. post-training time points?

The authors try to explain their negative brain-behavior correlation in the discussion (lower HC-MPFC connectivity at better MD across individuals), but I find their argument of "excessive integration (...) can disrupt specialized processing, leading to decreased performance" and further "heightened (...) connectivity might reflect an imbalance where excessive integration interferes with hippocampal function" far-fetched. While interpreting such results can be challenging, I expect empirical data/findings from animal studies/lesion data to back up such claims. This is the main result – please be as precise as possible.

"By using ... our study provided some causal evidence for the role of brain connectivity in MD" – this is a clear overstatement. The study does not show causal evidence for the role of brain connectivity; please rephrase this.

Minor points:

Please show individual data points / distributions in all plots.

Reviewer #2

(Remarks to the Author)

In this study, Iliopoulos and colleagues investigated event-related changes in functional connectivity associated with discriminating between similar images in a memory task, or mnemonic discrimination. Motivated by previous work, the authors focused primarily on connections between the medial temporal lobe (MTL) including the hippocampus, visual areas, and prefrontal cortex (PFC). The authors began by identifying three clusters of region-to-region connections that were modulated during successful identification of lures as distinct from a similar, previously seen image. Of these connectivity changes, hippocampal-PFC connectivity (in cluster 3) was negatively associated with mnemonic discrimination performance across individuals. A critical manipulation in this study was a two-week training paradigm followed by a session to assess changes in mnemonic discrimination performance and functional connectivity. The authors found a group-level effect of training on visual system connectivity, and within the training group, decreased hippocampal-PFC connectivity after training was associated with greater behavioral improvement on the in-scanner task.

The manuscript is clearly written, and the experimental approach is well-motivated. Understanding the interaction between MTL structures, the visual system, and prefrontal cortex is very relevant to memory research, and event-related functional connectivity via gPPI is a powerful tool for disambiguating the complex interactions that take place during memory. In particular, I found the difference in connectivity between correct rejection and repetition hit trials to be very compelling. However, I have reservations about the relationship between connectivity and individual performance differences, as well as the influence of the training manipulation. My concerns are broadly centered around a desire for additional methodological checks and a reduction of language asserting causality. Specific comments and suggestions below:

Major Suggestions:

1) Prior to any of the presented analyses, the authors performed clustering on the full 11 x 11 connectivity matrix, resulting in three significant clusters that were used for the rest of the study. While this approach is described briefly in the Methods, I believe the importance of this clustering approach warrants additional description in the Results, including details such as how connections are determined to be related, how many clusters were non-significant, what the connection-level threshold was for each cluster, and how many connections fell below that threshold in significant clusters.

2) Relatedly, the initial description and statistical testing of a negative correlation between hippocampal-PFC connectivity and lure performance is made on the level of single ROI pairs (Fig. 4). However, the correlation with out-of-scanner MST discriminability is performed using a composite connectivity score (Tables 3 and 4). Subsequent analyses use either ROI pair connectivity (Fig. 6) or the composite score (Fig. 7). The manuscript would benefit significantly from additional details about the motivation for using ROI- or composite-level correlations for certain analyses. Further, it would be valuable to see each of these analyses performed at both the ROI and composite level to note which effects persist.

3) The impact of the tested training paradigm on lure discriminability (Fig. 5 and Table 5) and LOC-OP connectivity (Fig. 6 and Table 6) are both individually convincing. However, lure discriminability and LOC-OP connectivity are not directly related, leaving open the possibility that these might be unrelated effects of the training paradigm. Correlating the delta in LOC-OP connectivity with the delta in discriminability across the training and control groups should address this concern. Absent this association, I would suggest the conclusion that the connectivity change reflects a refining of visual input to the hippocampus be tempered.

4) The observed negative correlation between the change in hippocampal-PFC functional connectivity and the change in discriminability (Fig. 7) is very interesting, but its relationship to the baseline results is significantly hindered by two features:

- First, this analysis appears to have only been performed in the training group. Given the lack of difference in hippocampal-PFC connectivity between the training and control groups, I wonder if the same negative correlation would also be present in the control group. If true, this might suggest that the training improves mnemonic discriminability through a mechanism distinct from hippocampal-PFC connectivity.
- Second, the lack of significant correlation between the change in hippocampal-PFC connectivity and the previously significant out-of-scanner MST task suggests this effect may not be fully comparable to the baseline effect. The authors address this discrepancy in the Discussion and propose a future longer intervention, but for this study I believe the conclusions from this analysis should be further tempered.

Minor Suggestions

5) A brief description of the additional, out-of-scanner memory tasks would be helpful in Results section 2.2.2.

6) It would be helpful to have further description in the Result or Discussion of why a significant correlation with the MST task, but not other out-of-scanner tasks, supports or doesn't support that hippocampal-PFC connectivity is involved in mnemonic discrimination.

7) It would be valuable to add a portion to the Discussion on how the ROIs and connectivity changes described in this paper map onto whole-brain MTL-associated functional connectivity networks identified during resting-state fixation.

Reviewer #3

(Remarks to the Author)
COMMSBIO-25-0619

The manuscript entitled “Hippocampal-cortical connectivity relates to inter-individual differences and training gains in distinguishing similar memories” by Iliopoulos, Güsten et al. describes an analysis of functional connectivity during a mnemonic discrimination task with objects and scenes. All participants in the experiment performed the mnemonic discrimination while undergoing fMRI scanning at two timepoints. Between the fMRI scanning sessions, half the participants were given training for mnemonic discrimination. Additionally, participants completed the MST as an independent measure of mnemonic discrimination performance. The authors examined functional connectivity for lure detection (correct rejection) trials contrasted with hit trials to create a “lure detection” (LD) contrast, which was used as the primary outcome variable for the analyses in the manuscript. The authors found decreased functional connectivity for lure correct rejections compared to hits in networks between MTL and occipital structures and increased connectivity between MTL and frontal regions. Interestingly, and counter-intuitively, they found that training resulted in decreases in the LD contrast between MTL and frontal regions.

1. My primary concern is that the LD contrast is difficult to interpret and is not theoretically motivated by the authors in the current manuscript. Previous papers examining functional connectivity in mnemonic discrimination tasks (e.g., <https://doi.org/10.1016/j.neuroimage.2017.01.062>) compared lure correct rejections to lure false alarms. This makes theoretical sense as lure correct rejections should involve pattern separation processes while lure false alarms would be driven more by pattern completion. However, it is difficult to interpret what a change in the present LD contrast would mean as correct recognition of targets seems theoretically distinct from behavior regarding lures. Further, it is not clear that the authors need to couch things in terms of a contrast as a functional connectivity score of 0 (i.e., no correlation) is meaningful in itself. I worry this contrast may lead to incorrect interpretations of the results; for example, in section 2.1 the authors state that there was “reduced connectivity” in the first cluster but from Figure 3B it is clear the effect is driven by increased connectivity for hits, which has been previously observed in the literature (e.g., <https://doi.org/10.1016/j.neuroimage.2012.07.056>).
2. More information is needed on the behavioral and statistical methods.
 - a. First, in section 2.1, the cluster-based inference method is not fully described. How are clusters defined?
 - b. How many trials were included per run in the 6-back mnemonic discrimination task?
 - c. Was there any indication given to the participants that the encoding block had ended and the retrieval block had begun, or was it treated as a continuous recognition task?
 - d. How was difficulty of mnemonic discrimination determined in the training task?
 - e. Is the training task appreciably different than the 6-back MD task? In other words, could participants be learning the task structure rather than learning mnemonic discrimination when undergoing the training intervention?
 - f. In section 4.4, the authors describe spatial smoothing as part of the fMRI data preprocessing but in the next paragraph (section 4.5.1) state that “unsmoothed” BOLD data were extracted from ROI’s. I found this confusing.
3. In a previous manuscript that presents a different analysis of these same data, the authors report behavioral performance differences between object and scene stimulus types. Considering that the ROIs involved in the current analyses are sensitive to objects vs scenes, the present analyses should include stimulus type as a factor. If the results do not change when stimulus type is a factor, then collapsing across stimulus types would be justified.
4. In section 2.2.2, the authors report: “Specifically, higher connectivity scores were associated with poorer performance on the lure discrimination index (LDI) and the corrected hit rate of the mnemonic similarity task, ... highlighting a targeted impact (Table 4). Conversely, no significant effects were observed for the other memory measures examined. This pattern underscores the specificity of connectivity effects to particular cognitive processes involved in MD.” However, the observation that functional connectivity was associated with both LDI and corrected recognition, which the authors in the introduction point out are often dissociable processes. That functional connectivity is associated with both measures weakens the argument for specificity of connectivity effects.
5. Minor: in section 2.2.2 the authors give the incorrect name of the MST (should be Mnemonic Similarity Task)

Version 1:

Reviewer comments:

Reviewer #1

(Remarks to the Author)

I thank the authors for their revisions. I still do not fully agree with their changes regarding comment 5: language such as “strong” or “compelling” evidence is redundant, if not misleading, as the approach is still only correlational and the authors did not perform Bayesian analysis that would allow conclusions regarding the strength of evidence, but I appreciate that the authors removed any reference to “causality” (as was also highlighted by reviewer 2). Thus, I’m happy to see this manuscript in print.

Reviewer #2

(Remarks to the Author)

The authors have addressed most of the concerns I raised in my original review, and I am very excited by their primary result that Hippocampal – PFC functional connectivity is related to inter-individual differences in mnemonic discrimination (MD). However, while the authors have performed important additional tests exploring functional connectivity in visual regions and training-related functional connectivity changes, I am concerned they have not fully integrated these results or my initial comments into their conclusions. I have outlined specific concerns below:

- 1) The authors find that after the MD behavioral intervention, individuals in the training group display stronger LOC-OP functional connectivity. However, additional analyses showed no significant correlation between LOC-OP and MD performance. Despite this, the authors have left seemingly unchanged their conclusion that the results “suggest that successful MD relates to a cortical network encompassing both lower and higher-order visual areas,” with “enhanced early visual top-down feedback” and “more fine-tuned input to the hippocampus.” I do not believe the added text indicating that their conclusions should be “considered with caution” is an adequate tempering, given the results. I suggest the authors rework the text to frame the visual results as an interesting association with the behavioral intervention, with more work needed to disambiguate if / how it relates to mnemonic discrimination.
- 2) The authors find that, in the training group alone, a reduction in Hippocampal - PFC functional connectivity after training is associated with improved performance on the in-scanner / trained MD task. I agree with the authors that the absence of this effect in the control group strengthens their result and addresses my previous concern (4a). However, I do not believe the removal of causal language sufficiently addresses the second part of my concern (4b). The baseline association between Hippocampal – PFC functional connectivity and the broader cognitive process of MD is compelling in large part because it generalizes to an MD task (the MST) beyond the one used to calculate the metric. That the post-training change in functional connectivity does not generalize in the same way suggests this effect may be task-specific, rather than related to MD broadly. The connection to MD generally is seemingly further weakened by the lack of a group-level difference in Hippocampal-PFC functional connectivity, a result the authors only mention superficially in the discussion. I suggest the authors further temper their conclusions from these results, highlighting the task-specific nature of the training-group effect and providing more extensive discussion of the null group-level effect and its implications.

Additional Minor Suggestions:

- The caption of Figure 4 describes that the Cluster 3 composite score only includes Hippocampal-PFC connections, rather than the entirety of Cluster 3 or only the significant connections within Cluster 3. This could be made clearer in the text, and it would be helpful to have further justification for why this approach was taken (instead of the two alternatives above).
- The introduction does a good job of providing background for the relationship between Hippocampal-PFC connections and MD, but would benefit from more unpacking and setup for the association with visual areas.

Reviewer #3

(Remarks to the Author)

I appreciate the authors' efforts to address my concerns but my primary concern remains, and if anything the additional information in the authors' reply strengthens my reservations about defining the LD contrast in terms of the contrast between hits and lure correct rejections. The heart of my concern is that there are different processes at work for target recognition compared to lure discrimination and these processes are differentially affected by the training program and the LD contrast confounds these potential differences. The arguments in the manuscript are about lure discrimination, so making the primary outcome measure specific to the neurocognitive computations underlying lure discrimination and not target recognition is important.

While it is true that the comparison between hits vs. correct rejections has a long history as a meaningful contrast in memory literature generally, for the specific question of mnemonic discrimination the comparison of correct vs. incorrect lure identification is a much more specific cognitive subtraction. The studies cited as justification for the LD contrast (refs 5, 8-10) do not seem to apply specifically to the present study. They did not assess functional connectivity (they all used event-based fMRI and/or behavioral measures to assess LD), so it is unclear if those approaches or results generalize to the present study. Further, behavioral results in those studies are different for hits than correct rejections (e.g., more than one of these studies show an effect of aging on lure false alarm rates but no effect on hit rates), indicating that there are likely different neurocognitive processes involved in correct target recognition compared to correct lure discrimination. The divergence in connectivity differences following training in the current study (e.g., the increase in connectivity for hits reported in section 2.1) further supports this interpretation.

I'm not convinced by the argument that the MST is not comparable to the current task for a couple reasons. First, the authors themselves include data indicating that their subjects' performance on the study-test version of the MST is highly correlated with performance in their main discrimination task. Second, there are a number of papers that used a continuous recognition version of the MST (e.g., Bakker et al. 2008) with similar results to the current task. Of note, the paper by Nash et al. (2015) used a continuous recognition version of the MST and observed very similar patterns of activation in the MTL for first presentations, hits, lure correct rejections and lure false alarms as the event-related version of the present task (Compare Gusten et al Figure 3 with Nash et al Figure 2). If all trials in the current study are “likely to trigger a pattern completion signal”, then it makes even more sense to use the comparison of successful pattern separation (lure correct rejections) vs unsuccessful pattern separation (lure false alarms) to isolate the effect of interest.

The behavioral data in Figure S1 indicate that lure performance is not at ceiling at baseline and the data presented in the Gusten paper indicates that lure performance improves but not to ceiling post training. Target performance, on the other hand, is very high at baseline, which may be one reason lure performance improved with training while target performance did not. Changes in connectivity for hits after training indicate that it is not necessarily a stable baseline.

Version 2:

Reviewer comments:

Reviewer #2

(Remarks to the Author)

I appreciate the authors' detailed responses to my remaining concerns, and their willingness to meaningfully incorporate both mine and other reviewer's feedback. I believe this has resulted in a stronger manuscript, and I support its publication.

Reviewer #3

(Remarks to the Author)

Thank you to the authors for supplying the two new analyses to address my concerns. I'm still not sure my larger theoretical issue is addressed in the revised manuscript, namely that the contrast of lure correct rejections vs hits does not isolate lure discrimination as cleanly as lure correct rejections vs lure false alarms. The new analyses provide some assurances that the results are not confounded by changes with target recognition, which strengthens the conclusions for me. I would like to have seen the new analyses in the manuscript proper than in supplemental materials, but that's not a major issue for me.

Reviewer #1 (Remarks to the Author):

The authors investigated fMRI-based functional connectivity and its relation to interindividual differences in memory discrimination (MD), and whether connectivity changes would be associated with MD improvements after cognitive training. Results showed a negative brain-behavior relationship: decreased hippocampal-prefrontal connectivity was linked to better MD performance across individuals and to larger improvements after cognitive training. Additionally, cognitive training increased functional connectivity between higher visual areas. The results are interesting, but major arguments in the introduction and discussion could be strengthened. I also have some concerns about their analytical approach. Overall, I hope these comments are helpful in improving the manuscript.

Major points:

1) I struggle to fully grasp the author's goal here: "An overarching question is whether neural resources explaining interindividual variability also explain training-related gains. If we find convergence (...) it would indicate that high performers and beneficiaries of cognitive training tap into similar neural resources". This appears trivial to me, but maybe I am missing the point. Why would you even assume a divergence between these aspects? The authors go on to explain what convergence / divergence would mean, but I struggle to see the point. Are there other examples from the literature where a potential divergence between interindividual variability and training-gains was reported?

Reply:

We thank the reviewer for raising this important point and for the opportunity to clarify our rationale. Our motivation for explicitly mentioning that we test this convergence stems from the fact that interindividual differences in task performance can arise from a variety of sources—not all of which are necessarily modifiable through training.

For example, individual differences in MD performance may reflect factors such as attention, perceptual processing, or even low-level neuromodulatory systems, some of which may not be amenable to change through cognitive training. In contrast, our intervention specifically targets mnemonic discrimination processes. It is possible that interindividual differences in mnemonic discrimination are related to interindividual differences in hippocampal circuitry. Cognitive training, however, may lead to improvements in mnemonic discrimination by modifying afferent and efferent connections of hippocampal circuitry, including the fidelity of visual representations. We have clarified this aspect in the introduction. Thus, training-related gains may reflect plastic changes in neural circuits that are not the primary source of baseline

variability. Such a potential divergence is relevant for targeting the suitable targets for cognitive enhancement.

To further clarify our rationale in the manuscript, we added the following section to our introduction in page 4:

“While it is possible that interindividual differences in MD are related to interindividual differences in hippocampal circuitry, cognitive training may lead to improvements in MD by modifying afferent and efferent connections of hippocampal circuitry, including the fidelity of visual representations”.

2) As long as the authors are not performing DCM or a variant thereof, I urge them to refrain from any comments on effective connectivity or directionality. gPPI is an extension of PPI, but it is essentially still a regression model, so, even if one defines a “seed” and a “target”, findings should be interpreted as increased/decreased functional connectivity between x and y, but not as effective connectivity from x to y. Similarly, while the precise connectivity profile clearly depends on the seed, the gPPI analysis will likely yield overlapping results when taking x or y as a seed. This is also visible in their results (e.g., Table 2), where many of the connections that are significantly linked to LD appear as two sides of the same coin. Considering this may help remove some of the redundancy and make the results more on point.

Reply:

We thank the reviewer for this point. Regarding directionality, we have removed all references to “effective connectivity” or directionality and now clearly state that our method provides a “functional connectivity” matrix.

For completeness and transparency, we have chosen to report each connection in its original seed–target format but we do not interpret these results as showing directionality. This approach allows readers to fully evaluate the data and generate hypotheses for future studies. This approach is also consistent with both our results and the theoretical framework of gPPI. For example, our findings show that training selectively affected the LOC-OP connection at the group level, but not the OP-LOC connection, highlighting the potential for asymmetry in the gPPI method (as also discussed in Masharipov et al., 2024).

In response to this, and to comment 2 from reviewer #2, we have revised the manuscript. We now provide in the main manuscript (section 2.2.1) the hippocampal-PFC composite

connectivity score linear model regarding the baseline connectivity–MD performance relationship (see Fig. 4).

Specifically, we now have included the following text:

“... Given that these hippocampal–PFC connections all exhibited a similar negative relationship with MD performance, we created a composite hippocampal–PFC connectivity score by averaging the z-scores of each connection. Using this composite score as the independent variable provided a more concise summary of the overall relationship. This model confirmed the strong negative association between LD connectivity and MD performance ($p = .001$, $b = -.032$; see Fig. 4)”.

Conversely, we now provide the individual ROI-pair analyses at the supplementary material instead (see Fig. S3). This change simplifies the presentation of the results for readers, addresses reviewers’ feedback in a synthetic fashion, and ensures consistency.

3) Head motion can substantially impact functional connectivity results and induce spurious findings. Did the framewise displacement differ between groups and between pre- vs. post-training time points?

Reply:

To ensure that motion did not differ systematically between groups or across sessions in our study, we conducted a mixed-design ANOVA on mean framewise displacement (FD), including group (training vs. control) and time point (pre- vs. post-training) as factors, with age and gender as covariates. As detailed in our new Supplementary Section S2.5 and illustrated in Supplementary Figure S7, there were no significant main effects of group ($F(1,49) = 0.03$, $p = .865$), time point ($F(1,49) = 1.63$, $p = .208$), nor a significant group \times time point interaction ($F(1,49) = 1.07$, $p = .307$) on mean FD. These results indicate that head motion was comparable between groups and across time points in our sample, supporting the validity of our functional connectivity analyses.

4) The authors try to explain their negative brain-behavior correlation in the discussion (lower HC-MPFC connectivity at better MD across individuals), but I find their argument of “excessive integration (...) can disrupt specialized processing, leading to decreased performance” and further “heightened (...) connectivity might reflect an imbalance where excessive integration interferes with hippocampal function” far-fetched. While interpreting

such results can be challenging, I expect empirical data/findings from animal studies/lesion data to back up such claims. This is the main result – please be as precise as possible.

Reply:

We thank the reviewer for this feedback. We ground our interpretation in the well-established framework of network integration and segregation, which posits that optimal cognitive performance depends on a dynamic balance between integration and local specialization (Shine et al., 2016; Shine & Poldrack, 2018; Tognoli & Kelso, 2014). We have clarified this in the discussion and have added additional references to several human neuroimaging studies and clinical populations showing that excessive integration (hyperconnectivity) can be linked to impaired cognitive function and reduced network efficiency, which includes brain pathological conditions.

For example, we also now cite recent evidence that hippocampal hyperconnectivity has been associated with poorer mnemonic discrimination in older adults, and with worse episodic memory in mild cognitive impairment. We have further strengthened our discussion as we now explicitly mention that our findings suggest that this principle of “over”-connectivity may be extended to healthy young adults and also task-connectivity data. Importantly, the link between hyperconnectivity and MD was found both at baseline and in the connectivity-change MD-change correlation following training, underlining the robustness of the effect.

Furthermore, we now also discuss the Inhibitory Deficit Theory which further strengthens the theoretical framework around the negative connectivity-behavior correlation. This theory posits that cognitive performance suffers when there is insufficient ability to filter out irrelevant or interfering information, which could occur when optimal segregation between neural circuits is not maintained. This provides a complementary cognitive framework for understanding how excessive hippocampal–PFC connectivity might disrupt the specialized processing required for successful mnemonic discrimination. This convergence of arguments from network neuroscience and cognitive theory lends support to our interpretation.

Our novel findings based on neuroimaging in human studies might generate new hypotheses and guide future mechanistic work in animal models. Our results may further motivate future causal and mechanistic research in animals and brain stimulation studies to directly test how hippocampal–PFC connectivity affects mnemonic discrimination. Therefore, we have now explicitly strengthened our discussion around this finding and made clear the need for future causal experiments in animals or brain stimulation.

For the revision of our manuscript to incorporate the above, please see page 21 of the Discussion for these changes. The revised discussion texts are also provided below:

“... Several human neuroimaging studies show that hyperconnectivity can be linked to impaired cognitive function and reduced network efficiency, a pattern also observed in several pathological brain states^{33,34}. For example, hippocampal hyperconnectivity has been associated with poorer MD performance in cognitively normal older adults³⁵ as well as worse memory performance in patients with mild cognitive impairment³⁶. Extending these findings, our results in young adults and specifically using MD-based task-connectivity data suggest that hyperconnectivity can be a marker of poorer memory performance even within healthy young populations, indicating that the integration-segregation principle may apply across the adult lifespan”.

“... A failure to maintain optimal segregation may allow non-specific or interfering influences to disrupt the processing required for successful discrimination. This interpretation aligns with the Inhibitory Deficit Theory, which suggests that cognitive performance suffers when there is insufficient ability to filter out irrelevant or interfering information³⁷. Network-level studies further support that optimal cognitive performance depends on dynamic shifts in brain organization, where both excessive integration and poor segregation can impair function^{32,38,39}. Thus, more connectivity is not always better, and optimal MD may require a fine-tuned balance between hippocampal specialization and its integration with prefrontal networks. Our results motivate future causal research in animals and brain stimulation where hippocampal-PFC connectivity can be directly manipulated to study these mechanisms”

5) “By using ... our study provided some causal evidence for the role of brain connectivity in MD” – this is a clear overstatement. The study does not show causal evidence for the role of brain connectivity; please rephrase this.

Reply:

We thank the reviewer for this comment. We partly agree. Associations observed in studies of interindividual variability cannot provide causal evidence. In order to go beyond the level of mere association, we investigated the convergence between interindividual variability and training-related gains in MD performance. We argue that this convergence is much closer to causality than mere association. However, we agree with the reviewer that proof for causality would require eliminating or impairing the connectivity pattern through direct manipulation.

In our original submission we had already highlighted this aspect: *“To strengthen causality, future research utilizing brain stimulation techniques can be helpful. For instance, applying inhibitory stimulation to the PFC during MD tasks could help determine whether disrupting PFC activity leads to improved performance in individuals who initially exhibit high hippocampal-PFC connectivity”*

Given all this, we have now revised the manuscript to use the term "compelling evidence" instead of “causal evidence”, while in the abstract we now mention “strong” instead of “causal” evidence.

Minor points:

Please show individual data points / distributions in all plots.

Reply:

Individual data points are shown in the majority of the plots, except in cases where the high number of connections would make the plots too crowded with data points and difficult to interpret. In these instances, we present bar plots with error bars to provide an estimate of the mean and variability (standard error) for each group or condition. This approach ensures clarity while still conveying the relevant statistical information to the reader.

Reviewer #2 (Remarks to the Author):

In this study, Iliopoulos and colleagues investigated event-related changes in functional connectivity associated with discriminating between similar images in a memory task, or mnemonic discrimination. Motivated by previous work, the authors focused primarily on connections between the medial temporal lobe (MTL) including the hippocampus, visual areas, and prefrontal cortex (PFC). The authors began by identifying three clusters of region-to-region connections that were modulated during successful identification of lures as distinct from a similar, previously seen image. Of these connectivity changes, hippocampal-PFC connectivity (in cluster 3) was negatively associated with mnemonic discrimination performance across individuals. A critical manipulation in this study was a two-week training paradigm followed by a session to assess changes in mnemonic discrimination performance and functional connectivity. The authors found a group-level effect of training on visual system connectivity, and within the training group, decreased

hippocampal-PFC connectivity after training was associated with greater behavioral improvement on the in-scanner task.

The manuscript is clearly written, and the experimental approach is well-motivated. Understanding the interaction between MTL structures, the visual system, and prefrontal cortex is very relevant to memory research, and event-related functional connectivity via gPPI is a powerful tool for disambiguating the complex interactions that take place during memory. In particular, I found the difference in connectivity between correct rejection and repetition hit trials to be very compelling. However, I have reservations about the relationship between connectivity and individual performance differences, as well as the influence of the training manipulation. My concerns are broadly centered around a desire for additional methodological checks and a reduction of language asserting causality. Specific comments and suggestions below:

Major Suggestions:

1) Prior to any of the presented analyses, the authors performed clustering on the full 11 x 11 connectivity matrix, resulting in three significant clusters that were used for the rest of the study. While this approach is described briefly in the Methods, I believe the importance of this clustering approach warrants additional description in the Results, including details such as how connections are determined to be related, how many clusters were non-significant, what the connection-level threshold was for each cluster, and how many connections fell below that threshold in significant clusters.

Reply:

We thank the reviewer for this comment. We provided additional description in the 2.1 results section addressing these points so that readers now can more easily understand the method from the results section. Specifically we adapted our manuscript adding the following:

“... using an a priori selection of 11 ROIs covering major prefrontal (PFC), medial temporal lobe (MTL), and visual areas, resulting in an 11 x 11 connectivity matrix (110 unique connections). To identify significant patterns of connectivity, we applied cluster-based inference using ROI-to-ROI network multivariate parametric statistics. This approach groups related connections into clusters based on their statistical association, and computes an F-statistic for each cluster. Significance was determined at the cluster level using FDR correction ($p < .05$, p -FDR). Within each significant cluster, a post-hoc connection-level

threshold of $p < .05$ (uncorrected) was applied to retain only the strongest contributing connections. Non-significant clusters and connections that did not meet the above threshold within significant clusters were not retained for further analysis.”

We also added to the 4.5.3 methods section further details and a more friendly description so that the used method is more accessible to non-expert readers, and we further explicitly cited the CONN’s documentation for readers interested in more details. The text added is the following:

“ ... Thus, connections were considered related if they formed part of a cluster that survived FDR correction at the cluster level, and only those connections within significant clusters that also met the connection-level threshold ($p < .05$ uncorrected) were retained for further analysis. This two-step thresholding approach ensured robust, statistically meaningful patterns of connectivity. For further details on the method, see the CONN toolbox documentation⁶³”.

Lastly, regarding the non-significant clusters no details are provided by the CONN toolbox, as those clusters are of no interest. Thus, those are not reported in the results.

2) Relatedly, the initial description and statistical testing of a negative correlation between hippocampal-PFC connectivity and lure performance is made on the level of single ROI pairs (Fig. 4). However, the correlation with out-of-scanner MST discriminability is performed using a composite connectivity score (Tables 3 and 4). Subsequent analyses use either ROI pair connectivity (Fig. 6) or the composite score (Fig. 7). The manuscript would benefit significantly from additional details about the motivation for using ROI- or composite-level correlations for certain analyses. Further, it would be valuable to see each of these analyses performed at both the ROI and composite level to note which effects persist.

Reply:

We thank the reviewer for this helpful comment. In response to this comment and to the comment 2 from reviewer #1, we have revised the manuscript to improve consistency and transparency regarding our use of composite versus individual ROI-pair connectivity analyses.

First, we now provide the composite connectivity score results for the baseline connectivity–MD performance relationship in the main text, in addition to the individual

ROI-pair analyses provided in the supplementary. This change simplifies the presentation for readers and ensures consistency across analyses. The text added is provided below:

“Using this composite score as the independent variable provided a more concise summary of the overall relationship. This model confirmed the strong negative association between LD connectivity and MD performance ($p = .001$, $b = -.032$; see Fig. 4)”.

Second, we have added more details in the Results sections 2.2.1 and 2.2.2 to clarify our rationale for aggregating these connections into a composite score. Specifically, we note that the hippocampal–PFC connections all exhibited a similar negative relationship with MD performance, justifying the use of a composite score to succinctly capture this effect. See for example:

“Given that these hippocampal–PFC connections all exhibited a similar negative relationship with MD performance, we created a composite hippocampal–PFC connectivity score by averaging the z-scores of each connection. ...”

Similarly, we added the following text in the results section 2.3.2, to clarify why we tested group differences for each individual connection:

“... Since training may have affected only certain connections, we tested a separate model for each connection identified in the baseline LD connectivity analysis to determine whether these were modulated by the training intervention (see Fig. 3). ...”

Furthermore, while addressing this point, we reviewed and refined the calculation of the z-scoring and averaging for the hippocampal-PFC connections that formed the composite connectivity score in the baseline fMRI data (pre-training). We have updated the statistics provided for its relationship with the performance in the additional out-of-scanner memory tasks (Table 4). Our conclusions for this analysis remain unchanged.

Finally, in line with the reviewer’s suggestion, we now provide supplementary figures (Figs. S3, S6) showing the results for each individual hippocampal–PFC connection, both for the baseline connectivity–MD performance relationship and for the connectivity change–behavior change analysis. This allows readers to examine the effects at both the composite and individual connection levels. Additionally we added Table S4 at the supplementary, which provides the cognitive training ANOVA results also for the hippocampal-PFC composite connectivity.

3) The impact of the tested training paradigm on lure discriminability (Fig. 5 and Table 5) and LOC-OP connectivity (Fig. 6 and Table 6) are both individually convincing. However, lure discriminability and LOC-OP connectivity are not directly related, leaving open the possibility that these might be unrelated effects of the training paradigm. Correlating the delta in LOC-OP connectivity with the delta in discriminability across the training and control groups should address this concern. Absent this association, I would suggest the conclusion that the connectivity change reflects a refining of visual input to the hippocampus be tempered.

Reply:

Thank you for this comment. We added the following part in the discussion to temper the conclusion and indicate caution:

“However, it is important to note that the physiological role of training-related LOC-OP visual connectivity changes for MD remains uncertain, given the absence of a significant correlation between changes in connectivity and changes in lure discriminability across participants (Fig. 7). Thus, our above interpretation should be considered with caution. Further research is needed to clarify the relationship between these neural and behavioral effects”

4) The observed negative correlation between the change in hippocampal-PFC functional connectivity and the change in discriminability (Fig. 7) is very interesting, but its relationship to the baseline results is significantly hindered by two features:

a. First, this analysis appears to have only been performed in the training group. Given the lack of difference in hippocampal-PFC connectivity between the training and control groups, I wonder if the same negative correlation would also be present in the control group. If true, this might suggest that the training improves mnemonic discriminability through a mechanism distinct from hippocampal-PFC connectivity.

Reply:

We thank the reviewer for this important comment. In response to this comment, we performed the analysis within the control group, and we found that this relationship is not significant within the control group. This is now reported in text in the results section 2.4 and the relevant plots are provided at the Supplementary (Fig. S5). This provides further support to our results.

b. Second, the lack of significant correlation between the change in hippocampal-PFC connectivity and the previously significant out-of-scanner MST task suggests this effect may not be fully comparable to the baseline effect. The authors address this discrepancy in the Discussion and propose a future longer intervention, but for this study I believe the conclusions from this analysis should be further tempered.

Reply:

Thank you for this comment. We have tempered the conclusion from this analysis, as we have replied to the reviewer's #1 Point 5.

More specifically, we have now revised the manuscript and use the term "compelling evidence" instead of "causal evidence" regarding the hippocampal-PFC connectivity, while in the abstract similarly we now mention "strong" instead of "causal" evidence. In addition, given that our results are significant within the training but not the control group, our data provides further support for our conclusions.

Minor Suggestions

5) A brief description of the additional, out-of-scanner memory tasks would be helpful in Results section 2.2.2.

Reply:

We thank the reviewer for this comment. We added the following brief description of the additional memory tasks in the Results section 2.2.2:

“In addition to the in-scanner MD task, participants completed several established memory tasks outside the scanner on a separate day. These included the Mnemonic Similarity Task (MST) to assess MD, the Rey-Osterrieth Complex Figure (ROCF) for visual memory and constructional ability, a modified verbal learning and memory test for verbal long-term memory, and the Object-in-Room Recall (ORR) task to evaluate spatial associative memory (see Methods for details). “

6) It would be helpful to have further description in the Result or Discussion of why a significant correlation with the MST task, but not other out-of-scanner tasks, supports or doesn't support that hippocampal-PFC connectivity is involved in mnemonic discrimination.

Reply:

We thank the reviewer for this suggestion. To address this point, we have expanded the discussion section to clarify why the observed significant relationship with the MST task, but not with other out-of-scanner memory tasks, supports the specificity of hippocampal–PFC connectivity for mnemonic discrimination. Specifically, we now note that hippocampal–PFC connectivity did not significantly relate to performance on other out-of-scanner memory tasks, such as the verbal learning, complex figure, or object-in-room recall tasks, which rely on broader memory or cognitive processes.

We further highlight that the association was evident for both metrics of the MD performance extracted from the MST task—the lure discrimination index (LDI) and the corrected hit rate—supporting the robustness and specificity of the finding. This specificity supports the interpretation that hippocampal–PFC connectivity is particularly relevant for the fine-grained discrimination processes required in MD, rather than reflecting a general association with broader memory or cognitive abilities. The revised text can be found in the Discussion (page 21-22). We also note that this clarification addresses a related point raised by reviewer #3 (point 4).

The relevant text added is the following:

“... Notably, hippocampal–PFC connectivity did not significantly relate to performance on other out-of-scanner memory tasks, such as the verbal learning, complex figure, or object-in-room recall tasks, which rely on broader memory or cognitive processes. This association was evident for both metrics of MD performance extracted from the MST task—the lure discrimination index (LDI) and the corrected hit rate—further supporting the robustness and specificity of the finding. This specificity supports the interpretation that hippocampal–PFC connectivity is particularly relevant for the fine-grained discrimination processes required in MD, rather than reflecting a general association with broader memory or cognitive abilities ...”

7) It would be valuable to add a portion to the Discussion on how the ROIs and connectivity changes described in this paper map onto whole-brain MTL-associated functional connectivity networks identified during resting-state fixation.

Reply:

We thank the reviewer for this thoughtful suggestion. While a detailed discussion of the relationship between task-based and resting-state MTL connectivity networks is beyond the

scope of the current manuscript, we have added a sentence to the Discussion highlighting this as an important direction for future research:

“Future studies could also examine how the task-based connectivity patterns identified here relate to whole-brain MTL-associated functional connectivity networks observed during resting-state, to further elucidate the generalizability and specificity of these findings.”

We believe this acknowledges the broader context and provides a valuable direction for future research, while maintaining the focus and readability of our manuscript.

Reviewer #3 (Remarks to the Author):

The manuscript entitled “Hippocampal-cortical connectivity relates to inter-individual differences and training gains in distinguishing similar memories” by Iliopoulos, Güsten et al. describes an analysis of functional connectivity during a mnemonic discrimination task with objects and scenes. All participants in the experiment performed the mnemonic discrimination while undergoing fMRI scanning at two timepoints. Between the fMRI scanning sessions, half the participants were given training for mnemonic discrimination. Additionally, participants completed the MST as an independent measure of mnemonic discrimination performance. The authors examined functional connectivity for lure detection (correct rejection) trials contrasted with hit trials to create a “lure detection” (LD) contrast, which was used as the primary outcome variable for the analyses in the manuscript. The authors found decreased functional connectivity for lure correct rejections compared to hits in networks between MTL and occipital structures and increased connectivity between MTL and frontal regions. Interestingly, and counter-intuitively, they found that training resulted in decreases in the LD contrast between MTL and frontal regions.

1. My primary concern is that the LD contrast is difficult to interpret and is not theoretically motivated by the authors in the current manuscript. Previous papers examining functional connectivity in mnemonic discrimination tasks (e.g., <https://doi.org/10.1016/j.neuroimage.2017.01.062>) compared lure correct rejections to lure false alarms. This makes theoretical sense as lure correct rejections should involve pattern separation processes while lure false alarms would be driven more by pattern completion. However, it is difficult to interpret what a change in the present LD contrast would mean as correct recognition of targets seems theoretically distinct from behavior

regarding lures. Further, it is not clear that the authors need to couch things in terms of a contrast as a functional connectivity score of 0 (i.e., no correlation) is meaningful in itself. I worry this contrast may lead to incorrect interpretations of the results; for example, in section 2.1 the authors state that there was “reduced connectivity” in the first cluster but from Figure 3B it is clear the effect is driven by increased connectivity for hits, which has been previously observed in the literature (e.g., <https://doi.org/10.1016/j.neuroimage.2012.07.056>).

Reply:

We thank the reviewer for raising this important point. While the correct lures versus false alarms contrast is indeed widely used, the correct lures minus repeats contrast we employed is also well-established and has been successfully applied in previous studies (e.g. Berron et al., 2018, 2019; Maass et al., 2019). This contrast builds on prior literature of correct recognition in new items versus repeats (eg. see the tradition of the late positivity potential, which has been based on the contrast of correct rejections versus repeats).

The rationale for our approach is now clarified in the introduction of the revised manuscript. Specifically, we have added the following text:

“... The LD contrast was preferred rather than focusing exclusively on the contrast of correctly versus incorrectly identified lures. The LD contrast provides a comprehensive assessment of the mechanisms involved in MD by capturing both explicit retrieval processes and implicit repetition-related neural responses. We contrast the brain response to the correctly identified lures versus the response to all the repeated stimuli, for which no discrimination is required. Notably, the LD contrast is well-established and widely used in paradigms where participants are presented exclusively with either repeated or lure stimuli^{5,8-10}, and has been shown to relate to both MD performance and biomarker data for neurodegeneration⁹. ...”

Furthermore, it is important to highlight that our paradigm differs structurally from the MST task used in the study referenced by the reviewer. The MST typically involves an incidental encoding phase followed by a delayed retrieval phase in the scanner, often including “foil” trials with entirely new stimuli. In contrast, our study uses a continuous recognition paradigm: participants view six images in a row (encoding phase), and after a short delay, are presented with six images again, each being either an exact repetition (repeat trial) or a slightly changed version (lure). Because the delay in our paradigm is short and there are no entirely new stimuli (i.e., no “foils”), all retrieval trials are likely to trigger a pattern completion signal due to the stimuli's high overlap and temporal proximity. This results in a high hit rate for repeats

and relatively few misses (see also Fig. S1). Given these differences, the correct lures minus repeats contrast is particularly well-suited for studying lure discrimination in our context.

An additional motivation for this contrast is that, in our paradigm, repeat trials serve as a robust baseline for pattern completion, while correct lure trials reflect mainly successful pattern separation. By contrasting correct lures with repeats, we can more specifically isolate the neural processes underlying mnemonic discrimination, rather than general task engagement or response frequency.

An additional methodological advantage of our approach is that, by using all repeat trials, we ensure a stable and consistent baseline across participants. Contrasts involving error trials (such as false alarms) can be confounded by variability in response patterns. Furthermore, our cognitive training intervention led to a significant increase in correct lures, but not in correct repeat trials, supporting the specificity of our findings to lure discrimination and associated connectivity. The validity of our contrast is further supported by the significant association between the LD functional connectivity and MD performance.

The rationale for focusing on contrast-based gPPI results is that gPPI as a method is primarily intended for use with contrasts. It is designed to test functional connectivity differences between experimental conditions by modeling the interaction between psychological variables (experimental conditions) and a physiological variable (imaging data) (McLaren et al., 2012). Therefore, while a gPPI model can include multiple conditions, the interpretation remains contrast-based: the beta values are meaningful in terms of differences between conditions, not as absolute connectivity strengths (Huang et al., 2024; McLaren et al., 2012). Given this justification, we focused our analyses on the LD contrast.

We also appreciate the reviewer's observation regarding the interpretation of "reduced connectivity" in section 2.1. We agree that, as with any contrast-based analysis, the observed effect can be driven by changes in either or both conditions. In this case, the effect is primarily due to increased connectivity during repeat (hit) trials, rather than decreased connectivity during lure correct rejections. We have revised the manuscript to clarify this point and to ensure our language accurately reflects the effect. Specifically, we have revised section 2.1 to reduce language ambiguity and clarify this point, now stating:

"The first cluster displayed lower connectivity in visual-to-visual and visual-MTL connections during lure detection, Specifically, connectivity for the repeat trials was higher than in the correct lures, driving this effect".

In summary, our contrast-based approach is both methodologically and theoretically justified in our study, given the structure of our paradigm, the properties of gPPI, and the observed behavioral effects — specifically, the selective improvement in lure discrimination.

2. More information is needed on the behavioral and statistical methods.

a. First, in section 2.1, the cluster-based inference method is not fully described. How are clusters defined?

Reply:

We thank the reviewer for this comment. We addressed this point by describing further the cluster-based inference method in the beginning of the 2.1 section. We replied in detail to this point raised also by Reviewer #2 point 1.

Specifically we adapted the 2.1 section of our manuscript by adding the following:

“... using an a priori selection of 11 ROIs covering major prefrontal (PFC), medial temporal lobe (MTL), and visual areas, resulting in an 11 x 11 connectivity matrix (110 unique connections). To identify significant patterns of connectivity, we applied cluster-based inference using ROI-to-ROI network multivariate parametric statistics. This approach groups related connections into clusters based on their statistical association, and computes an F-statistic for each cluster. Significance was determined at the cluster level using FDR correction ($p < .05$, p -FDR). Within each significant cluster, a post-hoc connection-level threshold of $p < .05$ (uncorrected) was applied to retain only the strongest contributing connections. Non-significant clusters and connections that did not meet the above threshold within significant clusters were not retained for further analysis.”

We also added to the 4.5.3 methods section further details and a more accessible description for readers. The text added is the following:

“ ... Thus, connections were considered related if they formed part of a cluster that survived FDR correction at the cluster level, and only those connections within significant clusters that also met the connection-level threshold ($p < .05$ uncorrected) were retained for further analysis. This two-step thresholding approach ensured robust, statistically meaningful patterns of connectivity. For further details on the method, see the CONN toolbox documentation⁶³”.

b. How many trials were included per run in the 6-back mnemonic discrimination task?

Reply:

As described in the 4.2.1 section, the task has 60 trials per run (15 trials per trial type x stimulus modality bin: lures-repeats x object-scenes), in addition to 20 baseline (scrambled trials). To make this more accessible from the results section already, we added a description in the first paragraph of section 2. Results:

“... The task consisted of two runs, each comprising 60 experimental trials -divided among lure and repeat conditions for both object and scene trial types- and 20 baseline perceptual control trials...”

c. Was there any indication given to the participants that the encoding block had ended and the retrieval block had begun, or was it treated as a continuous recognition task?

Reply:

It is a continuous recognition task, as we describe it in the 4.2.1 section. Therefore no indication was given to the participants that the encoding block had ended. To make it clear to readers from the results section already, we added additional information at the first paragraph of section 2. Results, now explicitly stating: *“... continuous recognition task (response options: “old” or “new”)...”*.

Additionally, we now also explicitly state this information at the methods 4.2.1 section: *“... while no indication is given to the participants that the encoding block has ended”*.

d. How was difficulty of mnemonic discrimination determined in the training task?

Reply:

The task difficulty (correct response rate) (i.e. discriminability) for each image has been assessed in a previous study (Güsten et al., 2024), as we describe in the 4.2.1 section. Therefore, the difficulty of discriminating each image has been pre-estimated in another study and this information has been used when setting up the training paradigm.

e. Is the training task appreciably different than the 6-back MD task? In other words, could

participants be learning the task structure rather than learning mnemonic discrimination when undergoing the training intervention?

Reply:

The training task is also based on the 6-back MD task, as explained in section 4.2.2. However it has an adaptive nature as subjects progress to the next level after they achieve a certain threshold of performance. In a separate experiment, reported in Güsten et al. (2024), it is shown that the adaptive nature of the applied training does lead to higher MD improvement as compared to a non-adaptive training format. This provides evidence that learning the task structure is unlikely to be the principal driver of training gains in our study.

f. In section 4.4, the authors describe spatial smoothing as part of the fMRI data preprocessing but in the next paragraph (section 4.5.1) state that “unsmoothed” BOLD data were extracted from ROI’s. I found this confusing.

Reply:

We thank the reviewer for this comment. We removed this phrase about “spatial smoothing” to avoid confusion, as this does not affect our analyses. The CONN toolbox performs spatial smoothing as part of the processing pipeline (e.g. standard practice for seed-to-voxel analyses), which however is not applicable for the ROI-to-ROI analyses we performed here. For ROI-to-ROI analyses, the unsmoothed BOLD data are extracted as correctly described in the manuscript.

3. In a previous manuscript that presents a different analysis of these same data, the authors report behavioral performance differences between object and scene stimulus types. Considering that the ROIs involved in the current analyses are sensitive to objects vs scenes, the present analyses should include stimulus type as a factor. If the results do not change when stimulus type is a factor, then collapsing across stimulus types would be justified.

Reply:

We thank the reviewer for this suggestion. We appreciate the importance of considering stimulus type, especially given the object- and scene-sensitivity of the ROIs involved. However, we would like to clarify that our current study was not designed to test stimulus-specific effects, nor do we make any claims regarding object- or scene-specific outcomes. The focus and hypotheses of our paper are centered on the general mechanisms of

memory discrimination and the impact of training, rather than on category-specific processing.

Importantly, the cognitive training intervention included both object and scene stimuli, and there was no a priori hypothesis or empirical evidence from our data to suggest that training would differentially affect one stimulus type over the other. In fact, as reported in our previous work (Güsten et al., 2024), the session \times group \times domain (stimulus type) interaction was not significant ($F(2, 155) = 0.3804, p = 0.6842$), indicating that the training effect was similar across domains. The main training effect persisted independently of stimulus type (session \times group: $F(2, 155) = 6.5184, p = 0.0019$). Thus, there is no evidence that the behavioral or neural effects of training were specific to either objects or scenes.

Additionally, our main findings regarding hippocampal–PFC interactions reflect higher-order cognitive processes that are not expected to be modulated by stimulus category, but rather by broader stimulus-independent mechanisms involved in memory discrimination and cognitive control. These processes are thought to operate beyond the level of stimulus-specific pathways.

Given these considerations, we believe that including stimulus type as a factor in our current analyses would shift the focus of the paper away from our central research questions and hypotheses. However, we agree that if future analyses or hypotheses specifically focus on category-specific effects, it would be appropriate to include stimulus type as a factor.

We have added a statement to the manuscript in the methods section 4.2.1 (page 26) to clarify this rationale and to note that the absence of a significant interaction with stimulus type justifies our approach of collapsing across categories in the present analyses. The text we added is the following:

“ It is important to note that although previous univariate analyses of these data have examined potential differences between object and scene stimulus types, the present study was not designed to test category-specific effects, and our hypotheses did not pertain to stimulus modality. Importantly, the cognitive training intervention included both object and scene stimuli, and prior analyses ¹⁰ found no significant interaction between stimulus type and training effects. Therefore, we collapsed across stimulus types in all analyses.”

4. In section 2.2.2, the authors report: “Specifically, higher connectivity scores were associated with poorer performance on the lure discrimination index (LDI) and the corrected hit rate of the mnemonic similarity task, ... highlighting a targeted impact (Table 4).

Conversely, no significant effects were observed for the other memory measures examined. This pattern underscores the specificity of connectivity effects to particular cognitive processes involved in MD.”

However, the observation that functional connectivity was associated with both LDI and corrected recognition, which the authors in the introduction point out are often dissociable processes. That functional connectivity is associated with both measures weakens the argument for specificity of connectivity effects.

Reply:

We thank the reviewer for this comment and the opportunity to clarify this point. As also explained in the 4.2.3 section, the corrected hit rate estimate ("old"|old - "old"|lure) is an alternative bias-corrected measure of discrimination. This means that it is a composite of both hits and false alarms and is not to be confused with just a measure of “recognition” (ie: hits/correct repeat trials). In our MD paradigm performed at the scanner, the A prime is computed in a similar manner (ie. by considering hits and false alarms). A major difference is that the MST task includes also the “foil” trials (completely unseen stimuli) which are not present in the two-alternative choice paradigms like our own (our paradigm has: “old”, “new” replies only; there is not a “similar” reply option as is the case in the MST task).

Therefore, our result does not weaken the argument for specificity of the connectivity effects, but they rather strengthen it. In other words, connectivity is associated with MD performance in another task using both the LDI and the corrected hit rate metrics of estimating the MD performance in this out-of-scanner task, which strengthens our specificity argument.

To make this point clear to the readers, we adapted the manuscript. We mentioned in the methods sections “4.2.3 Out-of-the scanner cognitive tasks”:

... “ *The corrected hit rate estimate (Pr) ("old"|old - "old"|lure) is an alternative bias-corrected measure of mnemonic discrimination performance, which is calculated based on the hits (correct repeats) minus the false alarms (incorrect lures) ” ...*

Moreover, we elaborate on this in the discussion section, as already mentioned in our reply to Reviewer’s #2 Point 6. The relevant section (page 21-22) is the following:

... “ *Notably, hippocampal–PFC connectivity did not significantly relate to performance on other out-of-scanner memory tasks, such as the verbal learning, complex figure, or object-in-room recall tasks, which rely on broader memory or cognitive processes. This association was evident for both metrics of MD performance extracted from the MST task—the lure discrimination index (LDI) and the corrected hit rate—further supporting the*

robustness and specificity of the finding. This specificity supports the interpretation that hippocampal–PFC connectivity is particularly relevant for the fine-grained discrimination processes required in MD, rather than reflecting a general association with broader memory or cognitive abilities. ” ...

5. Minor: in section 2.2.2 the authors give the incorrect name of the MST (should be Mnemonic Similarity Task)

Reply:

Thank you for this comment. We corrected this to avoid any possible confusion.

References

- Berron, D., Cardenas-Blanco, A., Bittner, D., Metzger, C. D., Spottke, A., Heneka, M. T., Fliessbach, K., Schneider, A., Teipel, S. J., Wagner, M., Speck, O., Jessen, F., & Düzel, E. (2019). Higher CSF Tau Levels Are Related to Hippocampal Hyperactivity and Object Mnemonic Discrimination in Older Adults. *The Journal of Neuroscience*, *39*(44), 8788–8797. <https://doi.org/10.1523/JNEUROSCI.1279-19.2019>
- Berron, D., Neumann, K., Maass, A., Schütze, H., Fliessbach, K., Kiven, V., Jessen, F., Sauvage, M., Kumaran, D., & Düzel, E. (2018). Age-related functional changes in domain-specific medial temporal lobe pathways. *Neurobiology of Aging*, *65*, 86–97. <https://doi.org/10.1016/j.neurobiolaging.2017.12.030>
- Güsten, J., Berron, D., Ziegler, G., & Düzel, E. (2024). *Increased recognition memory precision with decreased neural discrimination*. *Neuroscience*. <https://doi.org/10.1101/2024.06.19.599765>
- Huang, S., De Brigard, F., Cabeza, R., & Davis, S. W. (2024). Connectivity analyses for task-based fMRI. *Physics of Life Reviews*, *49*, 139–156. <https://doi.org/10.1016/j.plrev.2024.04.012>
- Maass, A., Berron, D., Harrison, T. M., Adams, J. N., La Joie, R., Baker, S., Mellinger, T., Bell, R.

- K., Swinnerton, K., Inglis, B., Rabinovici, G. D., Düzel, E., & Jagust, W. J. (2019). Alzheimer's pathology targets distinct memory networks in the ageing brain. *Brain*, *142*(8), 2492–2509. <https://doi.org/10.1093/brain/awz154>
- Masharipov, R., Knyazeva, I., Korotkov, A., Cherednichenko, D., & Kireev, M. (2024). Comparison of whole-brain task-modulated functional connectivity methods for fMRI task connectomics. *Communications Biology*, *7*(1). <https://doi.org/10.1038/s42003-024-07088-3>
- McLaren, D. G., Ries, M. L., Xu, G., & Johnson, S. C. (2012). A generalized form of context-dependent psychophysiological interactions (gPPI): A comparison to standard approaches. *NeuroImage*, *61*(4), 1277–1286. <https://doi.org/10.1016/j.neuroimage.2012.03.068>
- Shine, J. M., Bissett, P. G., Bell, P. T., Koyejo, O., Balsters, J. H., Gorgolewski, K. J., Moodie, C. A., & Poldrack, R. A. (2016). The Dynamics of Functional Brain Networks: Integrated Network States during Cognitive Task Performance. *Neuron*, *92*(2), 544–554. <https://doi.org/10.1016/j.neuron.2016.09.018>
- Shine, J. M., & Poldrack, R. A. (2018). Principles of dynamic network reconfiguration across diverse brain states. *NeuroImage*, *180*, 396–405. <https://doi.org/10.1016/j.neuroimage.2017.08.010>
- Tognoli, E., & Kelso, J. A. S. (2014). The Metastable Brain. *Neuron*, *81*(1), 35–48. <https://doi.org/10.1016/j.neuron.2013.12.022>

Reviewer #1 (Remarks to the Author):

I thank the authors for their revisions. I still do not fully agree with their changes regarding comment 5: language such as “strong” or “compelling” evidence is redundant, if not misleading, as the approach is still only correlational and the authors did not perform Bayesian analysis that would allow conclusions regarding the strength of evidence, but I appreciate that the authors removed any reference to “causality” (as was also highlighted by reviewer 2). Thus, I’m happy to see this manuscript in print.

Reviewer #2 (Remarks to the Author):

The authors have addressed most of the concerns I raised in my original review, and I am very excited by their primary result that Hippocampal – PFC functional connectivity is related to inter-individual differences in mnemonic discrimination (MD). However, while the authors have performed important additional tests exploring functional connectivity in visual regions and training-related functional connectivity changes, I am concerned they have not fully integrated these results or my initial comments into their conclusions. I have outlined specific concerns below:

1) The authors find that after the MD behavioral intervention, individuals in the training group display stronger LOC-OP functional connectivity. However, additional analyses showed no significant correlation between LOC-OP and MD performance. Despite this, the authors have left seemingly unchanged their conclusion that the results “suggest that successful MD relates to a cortical network encompassing both lower and higher-order visual areas,” with “enhanced early visual top-down feedback” and “more fine-tuned input to the hippocampus.” I do not believe the added text indicating that their conclusions should be “considered with caution” is an adequate tempering, given the results. I suggest the authors rework the text to frame the visual results as an interesting association with the behavioral intervention, with more work needed to disambiguate if / how it relates to mnemonic discrimination.

Our reply:

We thank the reviewer for this constructive comment. We have now revised the Discussion section accordingly to address this concern. Specifically:

1. **Reframed the Interpretation:** We have removed language that implied a direct causal link (e.g., "enhanced early visual top-down feedback"). We now frame the LOC-OP connectivity increase as a finding of training-induced plasticity, demonstrating that our intervention successfully modulated neural circuits within the visual system.

2. **Highlighting the Null Correlation:** We now bring the null correlation to the forefront of the interpretation, rather than mentioning it as a final caveat. We explicitly discuss this as an "interesting dissociation," where our intervention induced general plastic changes in visual networks, but only the changes in prefrontal-hippocampal connectivity were linked to individual differences in MD training gains. This reframing turns a potential limitation into a more nuanced discussion point of interest.
3. **Clarifying Future Directions:** We have moved the discussion of potential mechanisms (e.g., providing the hippocampus with more distinct inputs) to the end of the paragraph, clearly framing it as a hypothesis for future research to test, rather than a conclusion of the current study. To address the above points we revised our manuscript and further added the following second part of the visual-related paragraph (page 23):

"... However, the precise functional role of this change for MD improvement remains to be disambiguated, as we did not find a significant correlation between the magnitude of this LOC-OP connectivity change and individual gains in lure discrimination. This presents an interesting dissociation: while our intervention induced plastic changes in visual networks, only the changes in prefrontal-hippocampal connectivity were linked to individual differences in MD gains (Fig. 7). Future research is needed to clarify this relationship and test specific mechanisms, for example, whether enhanced visual representations facilitate downstream mnemonic processes by providing the hippocampus with more distinct inputs."

We believe these revisions, made in direct response to the reviewer's suggestion, have significantly improved the manuscript. The discussion is now more precise, better aligned with our data, strengthening the overall impact of our findings. We also made it more explicit which part of the discussion refers to the visual pre-training (baseline) connectivity results, and which refers to the training effects. We thank the reviewer again for their valuable feedback.

2) The authors find that, in the training group alone, a reduction in Hippocampal - PFC functional connectivity after training is associated with improved performance on the in-scanner / trained MD task. I agree with the authors that the absence of this effect in the control group strengthens their result and addresses my previous concern (4a). However, I do not believe the removal of causal language sufficiently addresses the second part of my concern (4b). The baseline association between Hippocampal – PFC functional connectivity and the broader cognitive process of MD is compelling in large part because it generalizes to an MD

task (the MST) beyond the one used to calculate the metric. That the post-training change in functional connectivity does not generalize in the same way suggests this effect may be task-specific, rather than related to MD broadly. The connection to MD generally is seemingly further weakened by the lack of a group-level difference in Hippocampal-PFC functional connectivity, a result the authors only mention superficially in the discussion. I suggest the authors further temper their conclusions from these results, highlighting the task-specific nature of the training-group effect and providing more extensive discussion of the null group-level effect and its implications.

Our reply:

We thank the reviewer for this insightful comment, which has helped us to improve our discussion. We have now substantially revised the Discussion section to address these points. Specifically:

1. **We now explicitly discuss the null group-level effect**, framing it as a common finding in intervention studies where individual differences in training response are significant (*"...This pattern—a significant brain-behavior correlation in the intervention group, in the absence of a group-level main effect—is common in intervention studies and suggests that individuals respond to training heterogeneously^{46–51}."*).

We highlight that our brain-behavior correlation is meaningful because it explains this inter-individual variability and identifies a neural marker of post-training MD gains.

2. **We now directly address the lack of transfer to the MST**, framing our result within the context of "near transfer" vs. "far transfer" effects, which is an established framework in the cognitive training literature (see p. 24):

"...

An important implication of our findings relates to the specificity of this training effect. The baseline association between hippocampal-PFC connectivity and MD generalized to an independent task (the MST). In contrast, the post-training change in connectivity was only associated with performance improvements on the trained task, not the MST. This suggests our intervention induced a task-specific "near transfer" effect, rather than a more generalized "far transfer" to another MD task. ... "

3. **Importantly, we now integrate these two points**, discussing how the *combination* of the null group-level effect and the task-specific brain-behavior correlation points to a coherent conclusion:

"... The lack of a group-level effect on connectivity, combined with the task-specific nature of the brain-behavior correlation, suggests that our two-week intervention was sufficient to modify the specific neural circuits engaged by the task, but perhaps not long or varied enough to induce a more global reorganization of the MD network that would generalize to untrained tasks. "

We believe this revised interpretation is more precise, better aligned with the broader cognitive training literature, and turns these limitations into an interesting and nuanced discussion of our findings. We thank the reviewer for encouraging us to explore these implications more fully.

Additional Minor Suggestions:

- The caption of Figure 4 describes that the Cluster 3 composite score only includes Hippocampal-PFC connections, rather than the entirety of Cluster 3 or only the significant connections within Cluster 3. This could be made clearer in the text, and it would be helpful to have further justification for why this approach was taken (instead of the two alternatives above).

Our reply:

We thank the reviewer for this helpful comment. As requested, we have revised the caption for Figure 4 to be more explicit about which connections were included. We now state:

...

"Figure 4. Hippocampal-PFC composite connectivity is negatively associated with MD performance. Brain connectivity-behavior linear regression model using all the significant hippocampal-PFC connections from cluster 3 of the LD analysis (Fig 3, Tables 1-2) as a composite z score. The X axis depicts the independent variable: LD composite PFC-HIPP connectivity values (Z-scores). This composite is the average of the six hippocampal-PFC connections' z score values. All these connections are part of cluster 3 of the LD analysis (6 out of the 7 connections in this cluster) and exhibited a significant negative link to MD. For a model fitted for each connection separately, see the supplementary Fig. S3. ... "

The detailed justification for this approach is also already provided in the main text (Section 2.2.1). To summarize, we created the composite from these specific connections because:

1. They constituted the vast majority of the cluster (6 of 7 connections) and were all hippocampal-prefrontal connections (the 7th connection is not; it is a within the PFC connection: IFG triangularis – IFG opercularis).
2. Importantly, they all exhibited a negative relationship with MD performance.

We believe the revised more detailed caption is now more clear to the reader. We also improved the clarity of the caption in several figures (for example see Figures 3-7, S3-S5), and we corrected a typo in the methods' figures (which show now consistently the right number of subjects for the training and control group).

- The introduction does a good job of providing background for the relationship between Hippocampal-PFC connections and MD, but would benefit from more unpacking and setup for the association with visual areas.

Our reply:

We thank the reviewer for this thoughtful suggestion. We agree that the visual system findings are an important component of our study. Our current approach achieves a more concise introduction and reserve the detailed interpretation of the visual findings for the Discussion section. We believe the introduction covers the visual areas enough, while the discussion section fully contextualizes these results after all other findings have been presented. While we appreciate the reviewer's perspective, we believe the current structure provides a good flow for the reader while achieving a good trade-off with the introduction length.

Reviewer #3 (Remarks to the Author):

I appreciate the authors' efforts to address my concerns but my primary concern remains, and if anything the additional information in the authors' reply strengthens my reservations about defining the LD contrast in terms of the contrast between hits and lure correct rejections. The heart of my concern is that there are different processes at work for target recognition compared to lure discrimination and these processes are differentially affected by the training program and the LD contrast confounds these potential differences. The arguments in the manuscript are about lure discrimination, so making the primary outcome measure specific to the neurocognitive computations underlying lure discrimination and not target recognition is important.

While it is true that the comparison between hits vs. correct rejections has a long history as a meaningful contrast in memory literature generally, for the specific question of mnemonic discrimination the comparison of correct vs. incorrect lure identification is a much more specific cognitive subtraction. The studies cited as justification for the LD contrast (refs 5, 8-10) do not seem to apply specifically to the present study. They did not assess functional connectivity (they all used event-based fMRI and/or behavioral measures to assess LD), so it is unclear if those approaches or results generalize to the present study. Further, behavioral results in those studies are different for hits than correct rejections (e.g., more than one of these studies show an effect of aging on lure false alarm rates but no effect on hit rates), indicating that there are likely different neurocognitive processes involved in correct target recognition compared to correct lure discrimination. The divergence in connectivity differences following training in the current study (e.g., the increase in connectivity for hits reported in section 2.1) further supports this interpretation.

I'm not convinced by the argument that the MST is not comparable to the current task for a couple reasons. First, the authors themselves include data indicating that their subjects' performance on the study-test version of the MST is highly correlated with performance in their main discrimination task. Second, there are a number of papers that used a continuous recognition version of the MST (e.g., Bakker et al. 2008) with similar results to the current task. Of note, the paper by Nash et al. (2015) used a continuous recognition version of the MST and observed very similar patterns of activation in the MTL for first presentations, hits, lure correct rejections and lure false alarms as the event-related version of the present task (Compare Gusten et al Figure 3 with Nash et al Figure 2). If all trials in the current study are "likely to trigger a pattern completion signal", then it makes even more sense to use the comparison of successful pattern separation (lure correct rejections) vs unsuccessful pattern separation (lure false alarms) to isolate the effect of interest.

The behavioral data in Figure S1 indicate that lure performance is not at ceiling at baseline and the data presented in the Gusten paper indicates that lure performance improves but not to ceiling post training. Target performance, on the other hand, is very high at baseline, which may be one reason lure performance improved with training while target performance did not. Changes in connectivity for hits after training indicate that it is not necessarily a stable baseline.

Our reply:

We thank Reviewer #3 for their feedback. We appreciate the opportunity to clarify our rationale and strengthen our manuscript. The reviewer's central concern is that the LD contrast confounds lure discrimination and target recognition, and that our training effects may be driven by changes in the repeat (hit) condition. We have taken these concerns seriously and performed two new, targeted analyses that empirically address the core of the concerns. These new results are now included in the Results section 2.5 and the Supplementary Information (Sections S2.6 and S2.7).

Theoretical validity of the LD contrast and its use with gPPI

The reviewer argues that the LD contrast is not well-motivated for connectivity and that prior activation-based fMRI studies do not apply to our study. We respectfully disagree. The LD contrast (correct lures minus repeats) has been used successfully in numerous prior task-fMRI studies of mnemonic discrimination, including those we cited. Our study uses an event-related design in the same manner as those previous fMRI studies. The gPPI method we used is tailored for event-related designs as it models how task events (conditions) modulate inter-regional connectivity. The core logic of contrasting psychological conditions to isolate a process of interest remains the same, whether the dependent variable is regional activation or task-modulated connectivity. Therefore, the theoretical justification for using an LD contrast from the event-related fMRI literature is directly applicable. Task-modulated connectivity is simply another way to model event-related fMRI data and it is thus a natural extension of activation-based studies.

Decomposing the LD training effects

We would like to respectfully clarify a potential point of confusion here. The reviewer states:

“The divergence in connectivity differences following training in the current study (e.g., the increase in connectivity for hits reported in section 2.1) further supports”

The results cited from our original Section 2.1 describe the *baseline, pre-training* connectivity patterns. They do not contain information on post-training changes. However, we find the reviewer's point about the stability of the repeat condition during training to be crucial.

To directly test whether our LD contrast is confounded by changes in the repeat condition, we decomposed the LD effect into its constituent conditions. We found that the training-related connectivity change was driven entirely by the correct-lure condition (see supplementary Section S2.6 and Fig. S8). The connectivity change for repeats alone did not differ from zero

($p = .863$), while the within-subjects difference of changes was highly significant ($p = .0001$). This empirically demonstrates that the repeat condition serves as a stable baseline in our training analysis, and the LD effect is driven by processes related to lure discrimination, not target recognition.

Sensitivity analysis using the correct vs. incorrect lures contrast

While we maintain that our a priori contrast is theoretically and methodologically sound, we believe that confirming our key finding with the reviewer's preferred contrast would significantly boost confidence in our results. To that end, we performed a sensitivity analysis using the correct lures vs. incorrect lures (false alarms) contrast. As detailed in our new Supplementary Section S2.7 and Figure S9, we replicated our main finding: greater decreases in hippocampal-PFC connectivity were significantly associated with larger improvements in MD within the training group ($p = .012$, partial $R^2 = .252$). This association was absent in the control group ($p = .327$, partial $R^2 = .042$). This demonstrates that our central conclusion about the role of hippocampal-PFC connectivity in training-related gains is robust and not dependent on the specific contrast used.

In summary, we believe our theoretical justification, combined with these two new data-driven analyses, provides compelling evidence for our approach. The decomposition analysis confirms the interpretability of our LD contrast training effects by showing that they are driven by correct lures, not repeats. The sensitivity analysis confirms the robustness of our central finding using the reviewer's suggested correct vs incorrect lures contrast. We have revised the manuscript and have included these new results to bolster confidence in our conclusions. We thank the reviewer again for this opportunity to further strengthen our manuscript.